



# Dynamical Connections between Large Marine Ecosystems of Austral South America based on numerical simulations

Karen Guihou[1-2-6], Alberto R. Piola[1-3], Elbio D. Palma[2-5], and Maria Paz Chidichimo[1-2-4]

1- Servicio de Hidrografía Naval, Buenos Aires

2 - Consejo Nacional de Investigaciones Científicas y Técnicas (CONICET)

3 - Departamento de Ciencias de la Atmósfera y los Océanos, Facultad de Ciencias Exactas y Naturales, Universidad de Buenos Aires, Argentina

4 - Instituto Franco-Argentino para el Estudio del Clima y sus Impactos (UMI-IFAECI/CNRS-CONICET-UBA), Buenos Aires, Argentina

5 - Departamento de Fı́sica, Universidad Nacional del Sur, Bahı́a Blanca, Argentina

- (current affiliation) NoLogin Consulting, Spain

**Correspondence:** Karen Guihou (karen.guihou@nologin.es)

**Abstract.**

The Humboldt Large Marine Ecosystem (HLME) and Patagonian Large Marine Ecosystem (PLME) are the two largest marine ecosystems of the Southern Hemisphere, respectively located along the Pacific and Atlantic coasts of southern South America. This work investigates the exchange between these two LMEs and its variability, employing numerical model results and offline particle tracking algorithms. 27 years of a $1/12°$ ROMS configuration (CMM) show a general poleward transport on the Southern region of HLME, and equatorward on the Patagonian Shelf (PS). A mean transport across Cape Horn's shelf ($68.1°W$) is 0.95 Sv.

Lagrangian simulations show that the majority of the southern PS waters originate from the upper layer in the southeast South Pacific (>200m), mainly from the southern Chile and Cape Horn shelves. The exchange takes place through Le Maire Strait, Magellan Strait, and the shelf-break. These inflows account to a net northeastward transport of 0.88 Sv at $51°S$ in the southern PLME. The transport across Magellan strait is small (0.1 Sv) but due to its relatively low salinity it impacts greatly the density and surface circulation of the coastal waters of the southern PLME. The water masses flowing into the Malvinas Embayment eventually reach the PLME through the Malvinas Shelf and occupy the outer part of the shelf. The seasonal and interannual variability of the transport are also addressed. On the southern PLME, the interannual variability of the shelf exchange is partly explained by the large-scale wind variability, which in turn is partly associated with the SAM index (r = 0.52).

## 1 Introduction

Large Marine Ecosystems (LMEs) are relatively large ocean regions that encompass coastal areas, from rivers and estuaries to the outer margins of continental shelves and major current systems, and are characterized by distinct bathymetry, hydrography, productivity, and trophically dependent populations (Duda and Sherman, 2002). The Humboldt Large Marine Ecosystem (HLME) and Patagonian Large Marine Ecosystem (PLME) are the two largest marine ecosystems of the Southern Hemisphere, respectively located along the Pacific and Atlantic coasts of South America (Fig. 1). The HLME hosts the world largest fisheries





and the PLME is one of the most productive regions in the Southern Hemisphere, with a wide variety of marine life (Falabella et al., 2009). As a result of its high primary productivity the PLME absorbs large quantities of Carbon dioxide (Bianchi et al., 2005; Kahl et al., 2017), accounting for around 1% of the global ocean net intake. More than 70% of the stock fisheries in these

LMEs are overfished or collapsed, while less than 1% of the region is protected (Heileman et al., 2009; Heileman, 2009). In their larval stages fish and other marine organisms are at the mercy of physical processes. At least some of the most important commercial species take advantage of high biological production at the shelves, and at the same time avoid being exported to the biologically poorer oceanic environment (Bakun, 1998). The larvae and plankton export mechanisms to the open ocean and their influence on the interannual variability in the recruitment of keystone species are unknown. Therefore an improved

knowledge of the spatial and temporal variability of exchange processes between these LMEs and with the deep ocean is essential to better understand, model and predict their future evolution in response to climate change and global warming, as anthropogenic activity and temperature rise are impacting the ecosystems all around the world (Halpern et al., 2008; Belkin, 2009).

The HLME is a prototypical eastern boundary upwelling system that extends from northern Peru to southern Chile in the

South Pacific Ocean, where it is adjacent to the PLME (Fig. 1). The large-scale circulation of the HLME includes the broad eastward flowing West Wind Drift at $\approx 43°$S that reaching the coast of South America splits into the equatorward Humboldt Current and the poleward flowing Cape Horn Current (CHC) (Strub et al., 1998). About 65% of the area of HLME corresponds to the Humboldt Current System and is under the influence of coastal upwelling from $4°$S to $40°$S. South of this latitude the HLME is mostly under the influence of downwelling-favorable (poleward) winds and the CHC that flows poleward along the

shelf break of the Southern Chilean Shelf (SCHS, $40-55°$S), a region with a complex fjord system. Further south, the shelf widens onto The Cape Horn Shelf region (CHS), marking the northern boundary of the Drake Passage.

In this region the westerlies are strong and schematic reconstructions of the annual mean ocean circulation at the southern tip of South America, based on hydrographic observations and models, show that the main flow patterns are fom the Pacific towards the Atlantic. The Antarctic Circumpolar Current (ACC) flows eastwards carrying cold and fresh (34 PSU) subantarctic

waters (Piola and Gordon, 1989; Peterson and Whitworth III, 1989), rich in nutrients (Acha et al., 2004; Romero et al., 2006) from the Drake Passage along the upper portions of the western slope of the Argentine Basin. In the Drake Passage the northern boundary of the ACC is marked by a salinity minimum north of the Subantarctic Front (SAF) (Peterson and Whitworth III, 1989; Kim and Orsi, 2014) (Fig. 1). Recent observational records with unprecedented spatial and temporal resolution in Drake Passage yielded an absolute ACC transport of $173.3 \pm 8.9$Sv (1Sv = $10^6$m/s) in Drake Passage (Chidichimo et al., 2014;

Donohue et al., 2016). The absolute ACC transport estimate is in agreement with Colin de Verdière and Ollitrault (2016) from a study combining Argo float displacements and historical hydrography from the World Ocean Atlas 2009 to determine a global mass conserving mean circulation, and with Firing et al. (2011) from a study using high-horizontal resolution directly measured shipborne ADCP velocity in Drake Passage.

The PLME is located in the Southwestern Atlantic Ocean and embraces the Patagonian Shelf (herafter PS), the largest

continental shelf of the Southern Hemisphere (Bisbal, 1995), stretching from $37°$S to $55°$S and reaching up to $850$km off mainland in its widest portion ($51°$S). The PS is characterized by low-salinity waters, fresher than 33.8 PSU, with minor



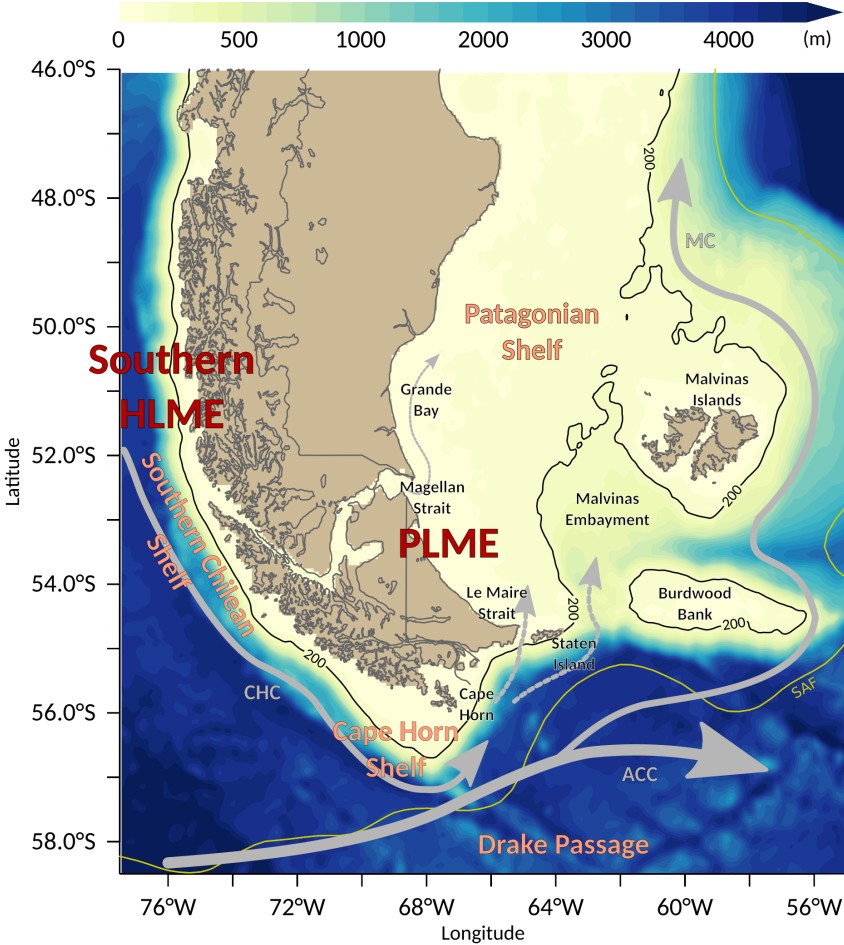

**Figure 1.** Domain of study and main geographic features. The colors show the bathymetry (m) of the ROMS model used in this study. The northern branch of the Antarctic Circumpolar Current (ACC), the Cape Horn Current (CHC), and the Malvinas Current (MC) paths are shown in grey and the Southern SubAntarctic Front (SAF), as defined by Kim and Orsi (2014), is shown in yellow.

seasonal and vertical salinity variations (Piola et al., 2010), and subjected to strong westerly winds and large amplitude tides (e.g. Palma et al. (2004)). An outstanding dynamical feature on the shelf is the Magellan Plume, derived from the discharge of relatively fresh waters through the Magellan Strait at $52.5°$S. This inflow explains the low salinity on the shelf and extends from $54°$S to $42°$S, that is up to $1800$km downstream (Piola et al. (2018) and references therein). The offshore boundary of the PS is marked by strong western-boundary currents, the Brazil Current at the northern extremity flowing southwards, and the Malvinas Current (herafter MC) south of about $38°$S, flowing northwards. The MC is derived from a deflection of the northern



branch of the ACC around the PS. The thermohaline contrast between the MC and shelf waters creates a remarkable shelf break front (e.g. Peterson and Whitworth III (1989); Acha et al. (2004); Saraceno et al. (2004); Romero et al. (2006)).

The Southern section of the PS is the area where the exchange with the Pacific and Subantartic waters take place, via the Magellan Strait, Le Maire Strait and the shelf break. The Magellan Strait links the Pacific and Atlantic at $52°$S, through a 570 km long channel of complex geomorphology, of varying width between a few km at the Primera Angostura and Segunda Angostura narrows and exceeding 50 km near the eastern mouth. The Magellan Strait is very shallow on the eastern basin (< 50 m), and exceeds 1000 m in the western basin. There is only one observationally based estimate Pacific to Atlantic net

throughflow via the Magellan Strait, ranging between 0.04 and 0.07 Sv (Brun et al., 2019), while numerical models indicate a net flow towards the Atlantic of 0.15 Sv (Palma and Matano, 2012). The 30-km wide Le Maire Strait separates the southeastern tip of Tierra del Fuego from Staten Islands (Isla de los Estados), at $54.83°$S. Situated downstream of Cape Horn, Le Maire Strait is under the influence of strong currents, flowing along the shelf break, and the SAF (Orsi et al., 1995). Little is known about the transport and dynamics into the strait itself. On the eastern side of Staten Island, the shelf break turns abruptly

north up to $51°$S where it diverges eastward towards the Malvinas Islands. The Malvinas Embayment (ME) separates the PS from the shallower Burdwood Bank (BB) and the open ocean.

The lack of direct transport observations make it difficult to determine the variability of the exchange between the HLME and PLME. The complex morphology of the Patagonian area (rugged coastline, narrow straits and enclosed bays) hampers the use of remote sensing such as satellite altimetry. Intrusions of high chlorophyll-a on the PS via Le Maire Strait have been

observed (Romero et al., 2006), but the cloud coverage make it difficult to use as timeseries. Surface drifters released in the Pacific usually end up ashore along the rugged Southern Chilean shelf, or follow the large-scale circulation paths (along the ACC and MC) but rarely reach the PS, and Argo floats are not designed to work on shelf studies. Hydrographic observations are too scarce to understand the time variability. The most notable published studies are the surveys of the Magellan strait by Panella et al. (1991), and the GEF Patagonia cruises which took place on the PS in 2005 and 2006 (Charo and Piola, 2014).

However such in-depth studies haven't been reconducted since and therefore cannot be used to evaluate the variability.

There have been a few modelling studies on the circulation of the PS evaluating its interactions with the deep ocean (e.g. Combes and Matano (2018)). However no studies have focused on the exchange between the SCHS and the southern PS and on their property exchanges with the deep ocean and their drivers. Here, using 27 years (1980-2006) of numerical simulations at $1/12°$, we aim to quantify the exchange between the LMEs and establish a budget of the volume transport around the

southern tip of South America. The flows through the straits and shelf-open ocean exchanges contributing to the variability of the transport and its link with climate indices are discussed. The paper is organized as follows: section 2 presents the datasets used in this study, section 3 presents the mean simulated transport in the region and assesses the exchange between the LMEs, section 4 focus on the seasonal and interannual variability of these exchange and the results are summarized and discussed in section 5.



## 2 Data

The model used in this work is the Regional Ocean Modeling System (ROMS, Shchepetkin and McWilliams (2005)) in a regional eddy-resolving configuration at $1/12°$. The model features a free surface, 40 $\sigma$-levels with terrain-following coordinates on the vertical (Shchepetkin and McWilliams, 2005) and is two-way nested in a parent grid of $1/4°$ of the Southern Hemisphere, allowing the forcing of the deep-ocean circulation at the boundaries. This model configuration has previously been described by Combes and Matano (2014a) and widely used for studies on the Patagonian Shelf (e.g. Combes and Matano (2014b, 2018)). CMM is forced at the surface by monthly ERA-Interim winds (Dee et al., 2011) and heat and freshwater fluxes from the Comprehensive Ocean-Atmosphere Data Set (COADS) (Da Silva and Young, 1994), with a tendency restoring term to the Pathfinder SST climatology. The main eight tidal components are also included. The bathymetry is smoothed from ETOPO1 (Amante and Eakins, 2009), to minimize the pressure gradient errors associated with terrain following coordinates (Mellor et al., 1994). The Magellan Strait is resolved as a channel of constant depth of 100m, with a fixed salinity of 31 at the Atlantic mouth to simulate the freshwater input from rivers and glaciers. For more specific details about the configuration, the reader is referred to Combes and Matano (2014a). The configuration's domain extends from $81°$W to $52.5°$W in longitude and $60°$S to $40°$S in latitude. 27 years (1980-2006) of monthly fields are used in this work, thus all sub-monthly and high-frequency processes such as tides and the synoptic atmospheric variability are filtered out.

In addition to the above-described simulation, the ORCA0083-N06 simulations based on the NEMO code (Madec, 2008), developed by the Marine System Modelling team at National Oceanography Centre (NOC), are used for comparison purpose. ORCA0083-N06 (hereafter referred to as ORCA) is a $1/12°$ global configuration, available monthly over 33 years (1978-2010). ORCA is a global configuration, featuring 75 vertical z-levels with a grid spacing ranging from 1m near the surface to 200m at the bottom, with partial steps. ORCA features a non-linear free surface. The bathymetry is derived from ETOPO2 (https://www.ngdc.noaa.gov/mgg/global/etopo2.html). ORCA uses a turbulent kinetic energy (TKE) mixing scheme (Blanke and Delecluse, 1993) and a total variance dissipation (TVD) advection scheme for active tracers (Madec, 2014). It is forced at the surface by the DRAKKAR forcing set atmospheric reanalysis (Brodeau et al., 2010), composed of ERA40 reanalysis winds and fluxes from the CORE2 reanalysis. ORCA does not include tides. It has successfully been used for artic (e.g. Kelly et al. (2018)) and Southern ocean (Mathiot et al., 2011; Duchez et al., 2014) studies.

Lagrangian analyses are conducted using the Ariane software (Blanke and Reynaud, 1997). This open source tool available at http://www.univ-brest.fr/lpo/ariane computes offline 3D streamlines from the simulated velocity field from an Ocean General Circulation Model (OGCM) such as NEMO or ROMS. It computes the time of advection between two cells, avoiding interpolation problems. There is no stranding parameterized. In this study Ariane is used in forward and backward modes applied to the 27 years of outputs from the ROMS configuration.



## 3 Exchange between the two LMEs

### 3.1 Mean Transport and Circulation

The PLME and HLME are subjected to various dynamical patterns. On the HLME, the narrow SCHS (extending from 0 to 100km width) that stretches from 48 to $53°$S, features a complex coastline and the western mouth of the Magellan Strait. It differs greatly from the CHS, at the tip of South America, where the continental slope orientation changes abruptly and the shelf is wider. In the Atlantic, the PLME can be separated in three sub-regions, featuring different dynamics and/or topography. First the Southern Patagonian Shelf (SPS), extending from the Southern tip of Tierra del Fuego to $47°$S, can be itself subdivided in 2 regions. SPS1, from Le Maire strait to Grande bay at $51°$S, presents the more complex topography, and is the region where the exchange with the Pacific waters takes place through the Magellan Strait, Le Maire Strait and the shelf-break. SPS2, located further north, extends from $49.5°$S to $47°$S. Further east, the transports are evaluated in the Malvinas Islands Shelf (MIS) region. The southern MIS is protected from a direct influence of the ACC by the shallower BB. Only a branch of this strong current, deflected between the PS and BB enters into the ME and flows along the Southern slope before joining the northwards flowing MC further north.

A volume transport budget of each of the above described areas is calculated over the period, as a mean over 27 years of the monthly velocities from CMM, vertically integrated up to 200-m deep. The transport is estimated normal to the 200-m isobath across the shelf break and is defined as positive poleward in the Pacific, equatorward in the Atlantic, and towards the shelf in the whole region. In the ME area, which is not a shelf section, the transport is calculated in the upper 200-m layer and defined positive in the direction of the mean flow, that is towards the north in the southern section and towards the East in the eastern section. Each region has a closed budget ($T_{net} \approx 0$). Fig. 2 shows the mean circulation in each of the previously defined areas, and table 1 presents the mean, standard deviation, minimum and maximum transports for each section. Black arrows schematically show the direction and intensity of the transport across the various sections. For comparison purpose, the mean transports from ORCA are also shown in Table 1.

On the Pacific shelf (SCHS), the current is poleward and then veers around the CHS, with an inflow from the shelf-break. The eastward transport at the Cape Horn section ($68°$W) reaches 0.95Sv, flowing in eastward on the northern continental shelf of Drake Passage. On the PS, the mean flow is positive (equatorward), and presents a total along-shelf transport of 1.33Sv at $47°$S.

In the SPS1 region, there is a net inflow from both Magellan Strait (0.10Sv), Le Maire Strait (0.17Sv) and the shelf break (0.62Sv). These inflows are compensated by a net northward outflow of 0.88Sv at $51°$S. This mass balance is in line with previous studies (0.5 to 0.8Sv, based on a regional implementation of the Princeton Ocean Model (Palma et al., 2008; Matano and Palma, 2008)). The transport through both straits accounts for 30% of the total inflow onto the SPS, with a larger inflow through Le Maire Strait than the Magellan Strait. The main source of waters to the SPS is through the shelf break (0.62Sv) and is associated with the inflow of waters across the southern ME section. Most of this on-shelf transport (0.5Sv) occurs south of $52.5°$S. Though the transport through the Magellan Strait is minor, it impacts greatly on the hydrographic characteristics of the PS, generating a coastal jet that leads to the development of a low-salinity plume extending downstream up to $46°$S. The MIS



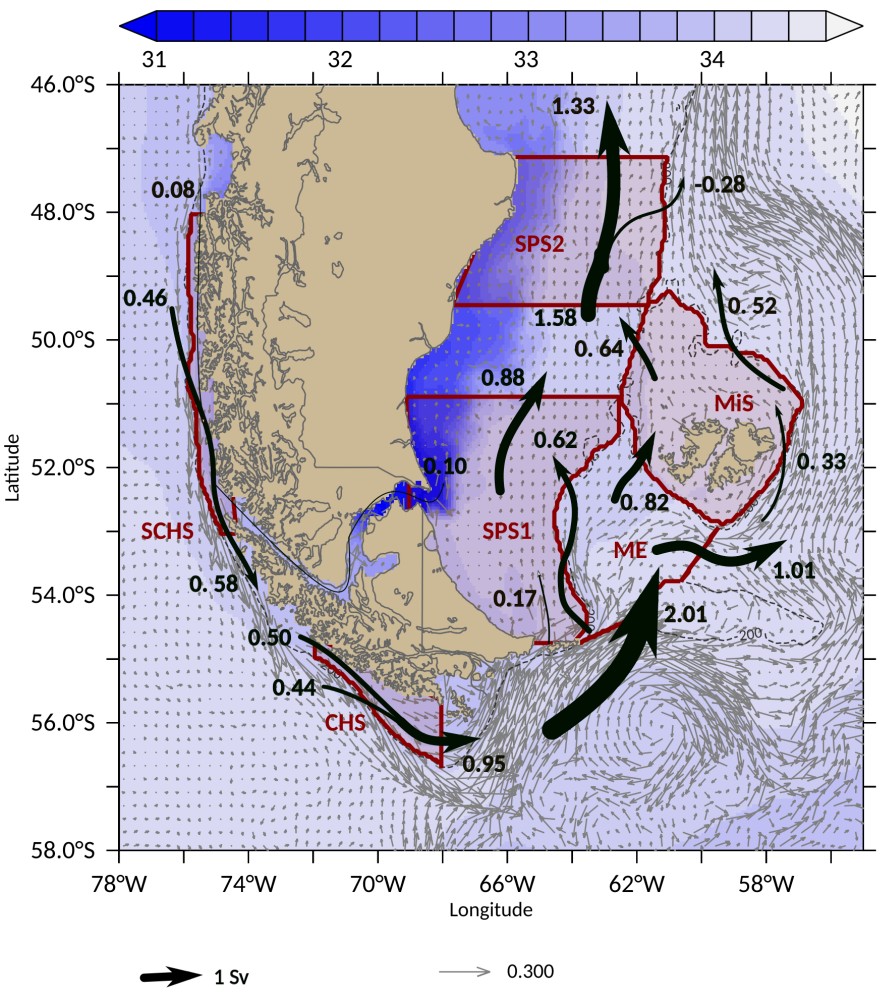

**Figure 2.** Mean total schematic transport over 27 years in the upper layer (0-200m) calculated from CMM, on the South Chilean Shelf (SCHS), Cape Horn Shelf (CHS), Southern Patagonian Shelf (SPS1 and SPS2), the Malvinas Islands Shelf (MIS), and Malvinas Embayment (ME). The colorbar represents the mean surface salinity from the model, and vectors the full mean depth-averaged velocity. Black arrows and values indicate the integrated transport across each section.

is more exposed to the influence of open-ocean waters and the MC. A net inflow of 1.15Sv penetrates through the southern section (mainly from the ME, accounting for 70% of the total inflow), and 0.52Sv exit northward towards the MC. A large fraction of the intruding waters (0.64Sv) remains on the shelf and flows westwards to join the net equatorward flow on the PS. Both flows, from SPS1 and MIS, join and flow into SPS2. This is the only location where we observe a significant offshelf (negative) transport (−0.28Sv) from the PS, even if most of the water (80%) entering the SPS2 through its southern boundary remains on the shelf and flows equatowards.





| Section | Mean | Std | Min | Max |
|---|---|---|---|---|
| **South Chilean Shelf (SCHS)** | | | | |
| Northern Section | 0.08 (0.02) | 0.08 | -0.15 | 0.40 |
| Shelf-Break | 0.46 (0.01) | 0.14 | 0 | 0.84 |
| Southern Section | 0.58 (0.07) | 0.12 | 0.18 | 0.86 |
| **Cape Horn Shelf (CHS)** | | | | |
| Western Section | 0.50 (0.25) | 0.14 | 0.14 | 0.92 |
| Shelf-break | 0.44 (0.48) | 0.14 | -0.05 | 0.97 |
| Eastern Section | 0.95 (0.78) | 0.21 | 0.37 | 1.58 |
| **Southern Patagonian Shelf (SPS1)** | | | | |
| Magellan Strait | 0.10 (0.08) | 0.03 | 0.03 | 0.22 |
| Le Maire Strait | 0.17 (0.52) | 0.04 | 0.04 | 0.28 |
| Shelf-Break | 0.62 (0.35) | 0.24 | 0.09 | 1.30 |
| Northern Section | 0.88 (0.90) | 0.23 | 0.31 | 1.56 |
| **Southern Patagonian Shelf (SPS2)** | | | | |
| Southern Section | 1.58 (1.07) | 0.35 | 0.73 | 2.52 |
| Shelf-break | -0.28 (-0.32) | 0.13 | -0.66 | 0.05 |
| Northern Section | 1.33 (0.83) | 0.28 | 0.63 | 1.94 |
| **Malvinas Embayment (ME)** | | | | |
| South | 2.01 (1.33) | 0.22 | 1.47 | 2.6 |
| East | 1.01 (1.08) | 0.28 | 0.3 | 1.88 |
| BB West | 0.25 (0.14) | 0.12 | -0.01 | 0.59 |
| **Malvinas Shelf (MIS)** | | | | |
| SouthWestern Section | 0.82 (0.35) | 0.16 | 0.42 | 1.58 |
| SouthEastern Section | 0.33 (0.10) | 0.09 | 0.09 | 0.61 |
| Northern Section | 0.52 (0.53) | 0.18 | 0.11 | 1.10 |
| Western Section | 0.64 (0.23) | 0.15 | 0.29 | 1.16 |

**Table 1.** Transport (in Sv) from CMM across the sections indicated in Fig. 2. Mean, Standard Deviation, Minimum and Maximum of the 27-year simulation. Mean values from ORCA are shown in brackets. Positive transports are poleward in the Pacific and equatorward in the Atlantic, and towards the shelf.

The above described transports are based on the numerical results from CMM. For comparison we have also analyzed the ORCA outputs over the same subregions. The mean flow calculated from the average velocity fields from ORCA is in good overall agreement with the results from the ROMS simulation (Table 1), as the circulation patterns are similar. The across-shelfbreak transports are smaller in ORCA. As a result, the inflows on the SCHS are limited, resulting in negligible transport along the Pacific shelf, but the total transport across the shelf section through CHS is of the same order as the CMM transport





(0.78Sv vs 0.95Sv). Likewise, the MIS region features a circulation pattern similar to CMM, but of smaller amplitude. The transport along the PS at $51°$S is of similar in both models (0.88Sv in COMM and 0.9Sv in ORCA). However, in contrast with the CMM results, the main source of waters to SPS1 in ORCA is the Le Maire Strait (0.52Sv), whereas only 0.35Sv enter via the shelf-break. On SPS2, the outflow towards the MC is of similar order in both simulations (-0.28 and -0.32 Sv).

### 3.2    Fate and origin of the LMEs waters

Lagrangian experiments are conducted to qualitatively investigate the fate of HLME and PLME waters. Using Ariane, particles are released in CMM's climatological velocity fields. Initially particles are released on all shelf grid points onshore from the 200-m isobath (see green areas in figures 3 a, 4 a and 5 a), at the surface, 50m and 100m depths, and advected by the mean flow during 90 days. It is important to note that though particles are released at a specific depth, they are free to displace vertically with the currents. Fig. 3, 4 and 5 show the initial/final positions of the particles released at the surface, at 50 m and 100 m

(panels a) and the trajectories, colored by depth (panels b). In order to render visibility to the figures, only one out of every 20 trajectory is represented in panels b.

On the SCHS, the particles are released from $47°$S along the Chilean coast up to Cape Horn at $68.3°$W (Fig. 3). The experiment represents a population of 1534 particles at the surface and 700 particles at 100-m deep. The similarity of trajectories at the 3 depths suggests that particles travel at similar velocities despite the different release depths. While a few particles enter

the Magellan Strait and flow directly towards the Atlantic, the majority take a poleward path to Cape Horn. The particles closest to the slope sink and join the CHC, at depths greater than 150m. Upon reaching Cape Horn, the particles that remain shallower than 100m (light blue and yellow trajectories in lower panels of Fig. 3 (b)) follow the shelf break and either intrude the PS via Le Maire Strait, or flow into the ME through the gap between Staten Island and BB. The particles deeper than 100m either follow a path along the SAF, around the southern slope of BB, or enter the ME along a path similar to the surface particles.

A similar experiment is carried out by releasing particles on the PS (Fig. 4). In this region, distinct trajectories are observed between the surface and subsurface layers. Most of the surface particles are rapidly advected towards the shelf-break, where they join the MC. When the MC reaches the Confluence zone at $\approx 40°$S, they retroflect and flow southwards towards the open-ocean (Fig. 4 (a)). All the surface particles remain at the surface. At 50 and 100 m depth, the particles drift onshore at relatively weak velocities. After 3 months of advection, most of the particles remain on the shelf. The particles close to the eastern mouth

of the Magellan Strait are advected upwards as they approach the inner shelf (Fig. 4 (b)). This onshore flow and subsequent upwelling compensates the eastward advection of the surface layer. Thus, the cross-shore circulation is consistent with Ekman dynamics under intense westerly winds, which drive the surface waters northeastward and induce the onshore flow in deeper layers and upwelling closer to the coast. In the central and eastern part of the shelf, the particles flow northward along the shelf. Though several particles reach the shelf-break very few are exported to the MC. These particles flow along the shelf-break, and

some are upwelled around $46°$S.

While these forward Lagrangian experiments confirm the flow of Pacific shelf waters into the PS and the equatorward flow in the Atlantic basin, they do not provide information about the possible inflow of open-ocean waters onto the shelf. To this end, we implemented an experiment prescribing a backward integration of 150 days from particles that reached SPS1 (Fig. 5).

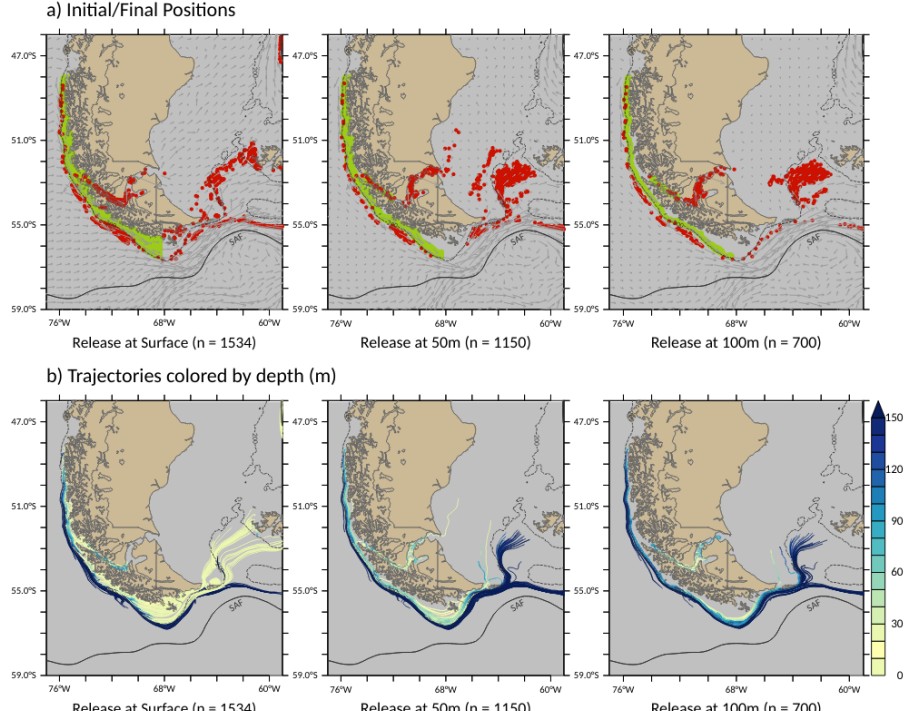

**Figure 3.** Lagrangian advection of particles during 90 days, released on the SCHS. Top panels (a): Initial positions (green) and final positions (red) of particles, released at 0 (left), 50 (middle) and 100 (right) meters. Vectors show the velocity at each depth. Bottom panels (b): particle trajectories colored by depth (in m) for initial release at 0 (left), 50 (middle) and 100 (right) m (1 out of 20 trajectories shown). The number of particles (n) for each experiment is shown and the SAF is indicated in black.

This experiment shows that the majority of surface waters in the PS originate in the Cape Horn shelf and enter the Atlantic through the Magellan and Le Maire straits and east of Staten Island. These particles are derived either from the surface waters in the southeast Pacific, north of 56°S at 76°W, or from the CHC. The deeper particles come from further north along the slope current (meaning they traveled faster) and several of these particles flow along the shelf break north of the SAF before reaching the CHS. It is important to note that apart from a few particles on the Malvinas Shelf, there are no waters originally deeper than 100m entering the shelf in this climatological experiment, neither are waters derived from south of the SAF reaching the

Atlantic shelf.

The above described particle paths indicate that the Drake Passage and CHS represent a key region for the exchange between the two LMEs, as no particle released south of 57°S in the averaged fields penetrates the PS. At the latitude of Cape Horn, the shelf features a change of orientation and is under the influence of strong flows associated with the SAF and the northern branch of the ACC. In order to better display the path of the particles flowing through the northern Drake Passage, quantitative

monthly experiments were conducted.





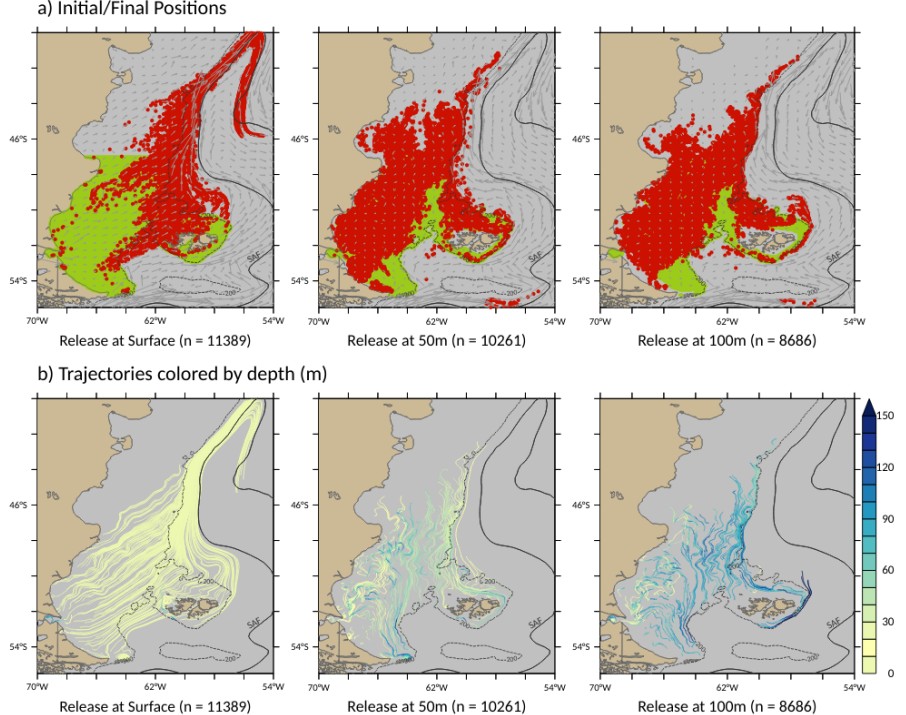

**Figure 4.** Lagrangian advection of particles during 90 days, released on the PS. Top panels (a): Initial positions (green) and final positions (red) of particles, released at 0 (left), 50 (middle) and 100 (right) meters. Vectors show the velocity at each depth. Bottom panels (b): particle trajectories colored by depth (in m) for initial release at 0 (left), 50 (middle) and 100 (right) m (1 out of 20 trajectories shown). The number of particles (n) for each experiment is shown and the SAF is indicated in black.

Particles are released on all grid points from 0 to 200m at $68.1°$W (Cape Horn), from the coast (at $55.7°$S) to $58.5°$S, and advected during one year by the monthly averaged currents and density fields from CMM. This simulation is repeated monthly from 1980 to 2005, that is 321 simulations of 365 days each. This corresponds to a release of 1649 particles per run (818 on CHS, that is from $55.7°$S to $56.67°$S, the remaining on the open ocean). At the end of each simulation, the particles which have reached the SPS1 or the ME are selected. The particles reaching the ME through the SPS1 are not counted twice, and are consequently only considered as particles reaching the SPS1. These specific particles are then run backwards from their initial location (along Cape Horn) during 30 days, to identify their origin. This experiment is designed to highlight the trajectory of the particles through the northern Drake Passage and CHS, from the Pacific to the PS, and assess their preferred path. The left panels in Fig. 6 display the cumulative density of particles (summed from all the simulations) flowing through the Drake Passage and reaching the SPS1 (Fig. 6 (a)) and the ME (Fig. 6 (b)), after 90 days of advection. The right panels in Fig. 6 show the propagation of the particles onto SPS1 (Fig. 6 (a)) and ME (Fig. 6 (b)) at 30, 90, 180 and 365 days.

On average for each simulation, 509 particles reach SPS1, and 381 reach the ME. Among them, 93% of the particles flowing through SPS1 and 49% of the particles flowing through the ME come from the CHS. No particle south of $57.48°$S reaches


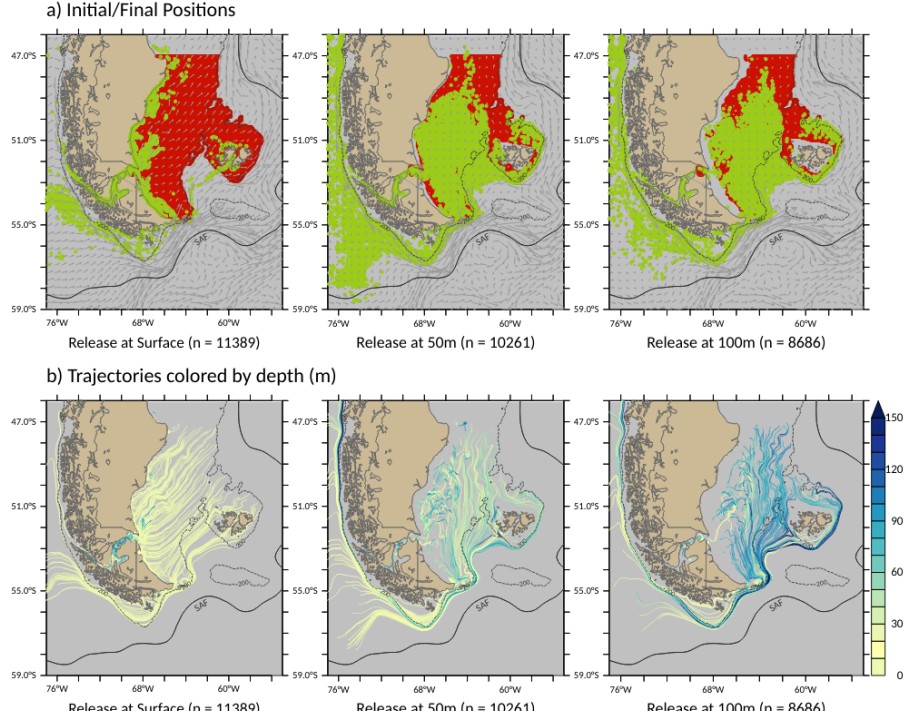

**Figure 5.** Backward Lagrangian advection of particles during 150 days, that reached the PS. Top panels (a): Initial positions (green) and final positions (red) of particles, released at 0 (left), 50 (middle) and 100 (right) meters. Vectors show the velocity at each depth. Bottom panels (b): particle trajectories colored by depth (in m) for initial release at 0 (left), 50 (middle) and 100 (right) m (1 out of 20 trajectories shown). The number of particles (n) for each experiment is shown and the SAF is indicated in black.

the SPS1. The density distribution shows that large number of particles enters the SPS1 through Le Maire Strait (Fig. 6 (a)).
These particles are advected all along the eastern side of the PS in 90 days, but they take longer to cover the coastal part of the shelf. After 1 year of advection, the particles have covered the entire SPS. The easternmost particles eventually flow south of the Malvinas Islands and join the MC. The particle flowing directly through the ME (Fig. 6 (b)) penetrate the MIS area rapidly (in 30 to 90 days) and enter SPS2, but do not reach the coastal zone during the time of these experiments. The majority of the flow joins the MC along the slope.

A similar experiment was carried out over the 200-1000m layer (Fig. 7): over the 137,172 particles that were released in total, only 127 reached SPS1, that is less than one particle per simulation. Moreover, the mean initial depth of these particles was 220m, and they were located very close to the shelf break (the most austral particle was initially at $56.94°S$). A relatively small number of particles reach the ME ($\approx 17$ particles per simulation). Some of these particles came from greater depths (up to 850m), but were mostly located close to the shelf break (90% of these particles were deployed north of $57°S$). This is
consistent with the previous result that the mayority of the waters occupying the PS are derived from the upper layer of the Pacific, and that little or no water from south of the SAF enters the PS.

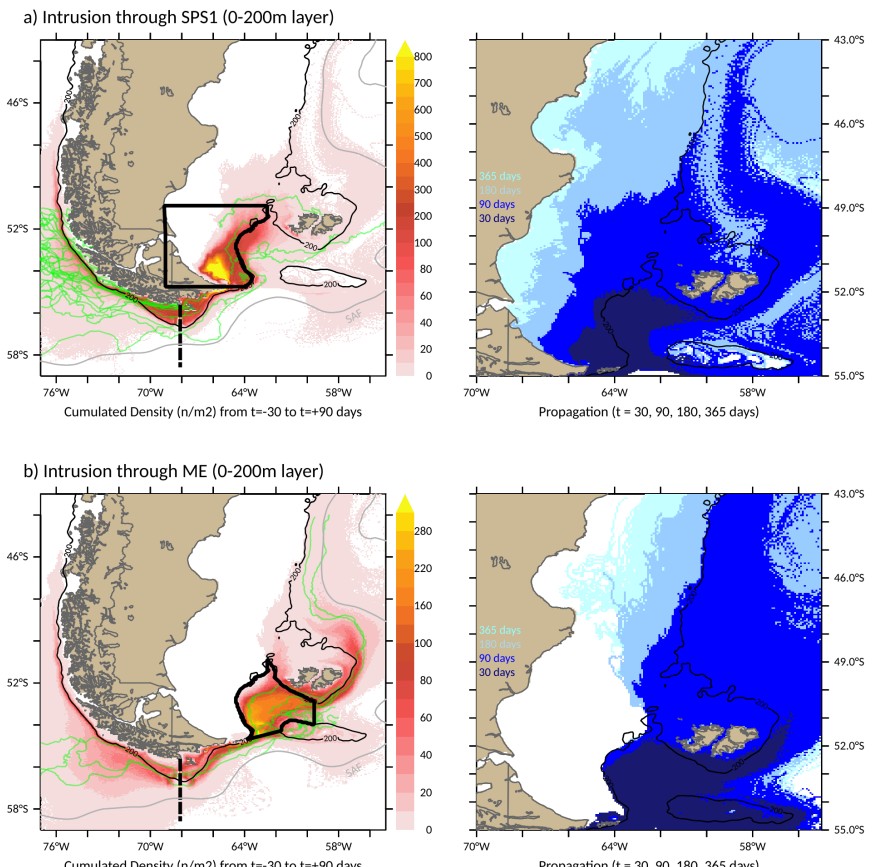

**Figure 6.** Pathways of monthly release of particles at Cape Horn ($68.1°$W, from $58.5°$S to the coast) on the 0-200m layer. Arrival on the Atlantic section through a) SPS1 and b) ME. Simulations are run backward for 30 days, and forward for up to 365 days. Left panels: cumulative squared-meter density (number of particles / $m^{-2}$), that is the cumulative spatial distribution after 324 simulations of 90 days. The drifter trajectories towards each area are shown in green. The dotted line at $68.1°$W shows the release section and the black area the SPS1 and ME areas respectively. Right panels: Color shading shows the propagation of the advected particles after 30, 90, 180 and 365 days, from dark to light blue.

Surface drifters trajectories, extracted from the Coriolis database (http://www.coriolis.eu.org/) are compared against the above described simulated trajectories (green lines in Fig. 6). Eighty-nine trajectories flowing on the HLME/PLME or through the northern Drake Passage, were recorded between 1980 and 2017. Among them, 33 drifters flowed onto the HLME or the PLME shelf. However, most drifters either ended washing ashore in the numerous fjords of southern Chile, or were released directly in the PLME, and therefore are not useful to portray the exchange between the two shelves. Only two drifters went through the Drake Passage and penetrated the PS briefly, before reaching the ME. Three additional drifters directly entered the ME after flowing through the Drake Passage. Even though the observational dataset is small, the observed and simulated


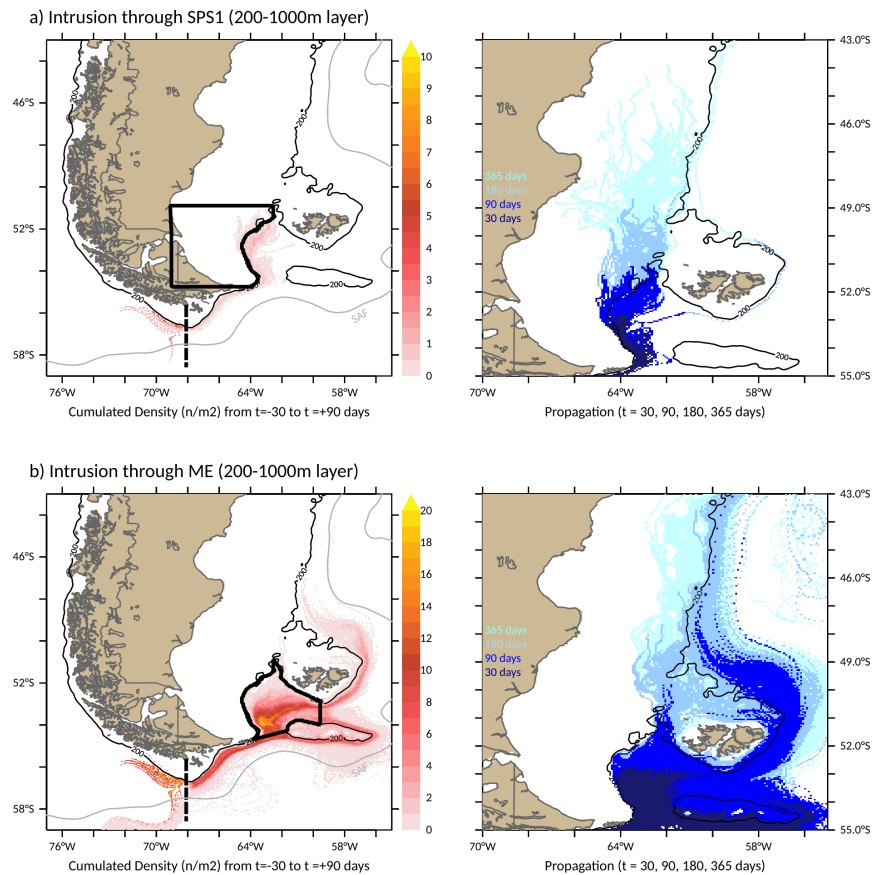

**Figure 7.** Pathways of monthly release of particles at Cape Horn (68.1°W, from 58.5°S to the coast) on the 200-1000m layer. Arrival on the Atlantic section through a) SPS1 and b) ME. Simulations are run backward for 30 days, and forward for up to 365 days. Left panels: cumulative squared-meter density (number of particles / $m^{-2}$), that is the cumulative spatial distribution after 324 simulations of 90 days. The dotted line at 68.1°W shows the release section and the black area the SPS1 and ME areas respectively. Right panels: Colors show the propagation of the advected particles after 30, 90, 180 and 365 days, from dark to light blue.

trajectories are in good qualitative agreement with the model lagrangian experiments, as no drifters flowing through the Drake
Passage south of the SAF entered the embayment. This supports our conclusion about the Pacific origin of the PS water masses.

## 4   Variability of the exchange

This section assesses the seasonal to interanual variability of the transport through selected sections, and investigates the origin of this variability.





### 4.1 Seasonal variability

On the southern HLME, the seasonal across-shelf transport onto the SCHS is lowest in June (green line of Fig. 8 (a)), coinciding with a minimum in the zonal wind seasonal variability. At the same time, the increase in southward transport through the northern shelf section (blue line of Fig. 8 (a)) results in a small change of the southward export across the southern boundary of SCHS (0.16 Sv, red line in Fig. 8 (a)). A similar seasonal cycle is found across the shelf of CHS and at its eastern section (Fig. 8 (b)), but presents a higher amplitude ($\approx 0.35$Sv) with a peak in austral summer and a minimum in June.

While the seasonal cycles of wind and transport intensity are "correlated" on the SCHS and CHS, this is not the case on the PS, where the transport minimum is observed in austral spring (Fig. 8, (c) to (e)) and the averaged wind on SPS1 presents a lower (and negative) correlation with the across-shelf transport at that location. While the transport through the Magellan and Le Maire straits presents lower variability, the transport across the northern boundary of SPS1 is highly correlated with the transport variability across the shelf-break section (r = 0.95), with a minimum in October (0.75Sv) and a maximum in summer

(1Sv). This can partly be explained by an increased offshore Ekman surface transport in late summer, which decreases the inflow from the shelf-break. Combes and Matano (2018) argue that these seasonal variations are modulated by the ACC. They also indicate that variations in the shelf inflow (i.e. through SPS1 shelf break) and the ACC transport are out of phase. Thus, an increase (decrease) of the latter produces a weakening (strengthening) of the former. The larger ACC transport during late winter augments its inertial tendency to flow eastward along the southern flank of the BB and, therefore, weakens the portion

of the flow diverted into the ME (Fetter and Matano, 2008). In summer a weaker ACC increase its interaction with the ME boosting a larger inflow to SPS1.

The transport variability on the MIS is more complex. The incoming transport through the southern boundary (blue line in Fig. 8 (e)) does not present a marked seasonal variability. However, the northward flow towards the MC increases in austral winter (red line in Fig. 8 (e)), which is partially compensated by a transport decrease towards the SPS (green line in Fig. 8

(e)). The transport through the southern and northern boundaries of SPS2 (blue and red lines in Fig. 8 (d), respectively) have very similar seasonal cycles to those observed at the offshore boundaries of SPS1 and MIS (green lines in Fig. 8 (c) and 8 (e), respectively). Since the transports through the Magellan and Le Maire straits are relatively small, the seasonal variability of the northward transport in the southern PS appears closely associated with the variability of the shelf-open ocean exchanges.

### 4.2 Interanual Variability

Fig. 9 shows the timeseries over 27 years of the transport anomaly along the borders of the key areas from the CMM simulations. Also shown in Fig. 9 are the spatially averaged zonal winds from ERA-Interim in the same areas, used as forcing for the simulations. A 12-month running mean is applied to the data to filter the seasonal signal. No significant trend in the transport is observed over the nearly 3 decades of simulations, neither in the forcing. The interannual variability of the transport anomalies in ORCA is in very good agreement with CMM (not shown), with a correlation above 0.7 on CHS and PS.

On the SCHS, the most significant anomaly occurs in 1998, when the inflow from the deep ocean decreased by $\approx 0.2$ Sv, associated with a low zonal wind stress intensity. Further downstream, the southward inflow to the CHS from the shelf


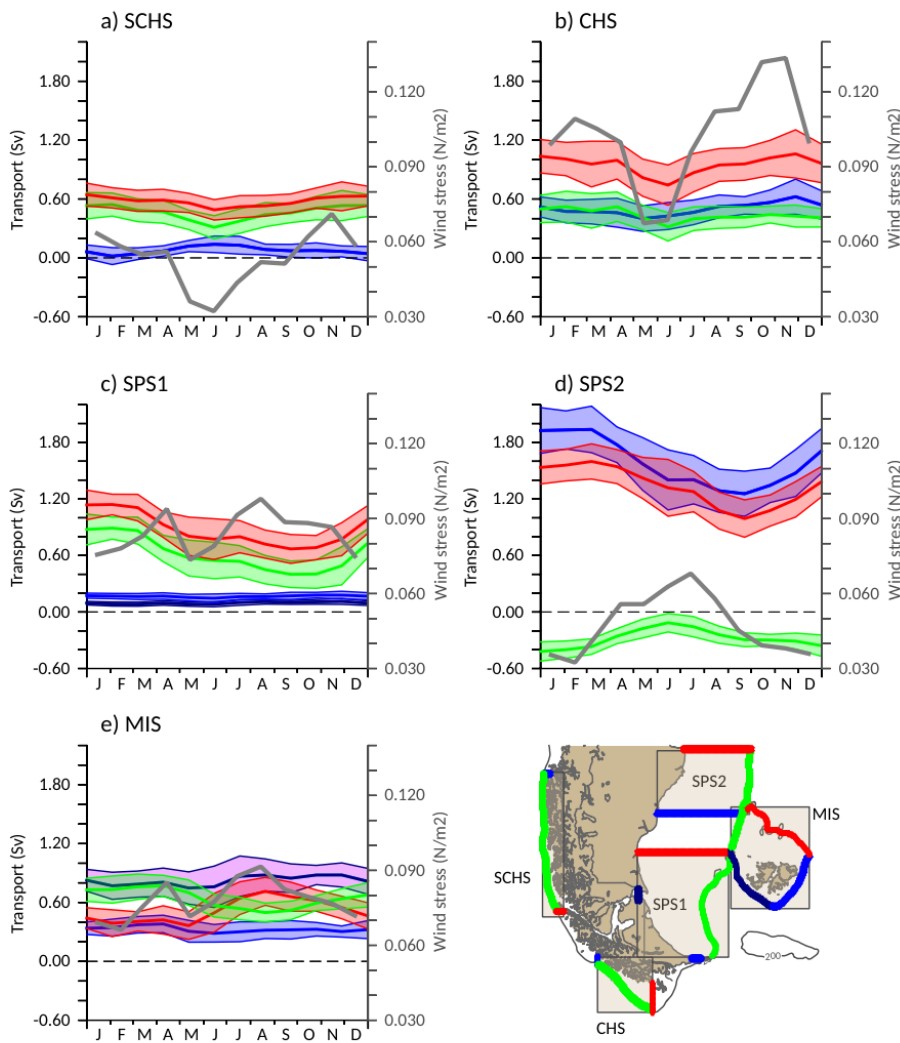

**Figure 8.** Mean seasonal variability of the transport across the key areas: SCHS (a), CHS (b), SPS1 (c), SPS2 (d), MIS (e). The transport is defined as positive in the direction of the main flux (polewards in the Pacific, Equatowards in the Atlantic) and towards the shelf. The shaded areas represent the standard deviation from the 27 years of simulation. Bottom left panel shows a reference map of sections of each key area where the seasonal transport has been calculated and their associated colors. The grey line in each panel is the zonal wind stress in the area, calculated in the shaded regions indicated in the reference map.

further north (blue line in Fig. 9 (b)) is decorrelated from the exchange across the 200-m isobath, but the resulting eastward transport at 68°W (red line in Fig. 9 (b)) depends on both the along-shelf and across-shelf variability (correlations of 0.72 and 0.79 respectively). Both regions present an interannual variability correlated with the variability of the zonal and meridional components of the local wind (Table 3).






The variability of the transport anomaly on the southern PLME (SPS1 and SPS2) presents larger amplitudes than on the southern HLME, with remarkable minima in 1989 and 1999, and maxima in 1996 and 2003. The inflow through Le Maire Strait, although small compared to the across-shelf exchange, is highly dependent on the upstream variations over the CHS (r = 0.79, Table 2). The transport across the southwestern MIS section (cyan line in Fig. 9 (e)) and the SPS Shelf-break sections (green lines in Fig. 9 (c) and (d)), though both are bounded by the ME, are poorly correlated (r = 0.14) suggesting that these exchanges are controled by local dynamics. Indeed the interanual anomaly of transport across the southwestern section of MIS if highly correlated to the interanual zonal wind averaged over the MIS area (r = 0.74, Table 3). On the other hand, the intrusion of the MIS waters onto the PS through its western section are moderately correlated with the SPS1 transport anomalies (r = 0.37), suggesting different drivers. The transport variations across the southern boundary of SPS2 (blue line in Fig. 9 (d)) are equally dependent on the variability of the northern boundary of SPS1 (red line in Fig. 9 (e)) and the western boundary in MIS (r = 0.82 and 0.81, respectively). Positive transport anomalies across the southern boundary of the SPS2 are associated with increased outflow at the shelf-break.

| | MIS (West) | MIS (South-West) | SPS2 (South) | SPS1 (Shelf) | SPS1 (LMS) | CHS (East) |
|---|---|---|---|---|---|---|
| SCHS (South) | 0.12 | 0.49 | -0.18 | -0.52 | 0.40 | 0.65 |
| CHS (East) | 0.13 | 0.43 | -0.08 | -0.45 | **0.79** | |
| SPS1 (LMS) | 0.17 | 0.50 | 0.14 | -0.18 | | |
| SPS1 (Shelf) | 0.37 | 0.14 | **0.82** | | | |
| SPS2 (South) | **0.81** | 0.52 | | | | |
| MIS (SouthWest) | 0.57 | | | | | |

**Table 2.** Correlation between interanual transport anomalies from CMM across selected sections: SCHS (South), CHS (East), SPS1 (LMS and Shelf), SPS2 (South), MIS (South-West and West). High-correlations (equal or above 0.7) are shown in bold.

| | SCHS (south) | CHS (East) | SPS1 (LMS) | SPS1 (Shelf) | SPS2 (South) | MIS (South-West) | MIS (West) |
|---|---|---|---|---|---|---|---|
| SAM | 0.38 | 0.50 | 0.17 | -0.52 | -0.28 | 0.07 | 0.03 |
| Local Zonal Wind | **0.73** | **0.75** | **0.71** | -0.41 | 0.23 | **0.74** | 0.27 |
| Local Merid. Wind | **-0.77** | -0.58 | 0.03 | 0.68 | 0.38 | -0.25 | 0.16 |

**Table 3.** Correlation between the SAM, local wind anomalies, and the transport anomalies across selected sections calculated from CMM: SCHS (South), CHS (East), SPS1 (LMS and Shelf), SPS2 (South), MIS (South-West and West). For each section, the local wind anomaly is calculated in the shaded regions indicated in Fig. 9. High-correlations (equal or above 0.7) are shown in bold.

The analysis of the spatio-temporal variability of the simulated transport suggests that the interannual transport variability at shelf-break boundary in SPS1 (green line in Fig. 9 (c)) controls a substantial part of the variability of the northward transport all along the PS shelf. To further understand the nature of the interannual transport variability on the PS we compare the transport





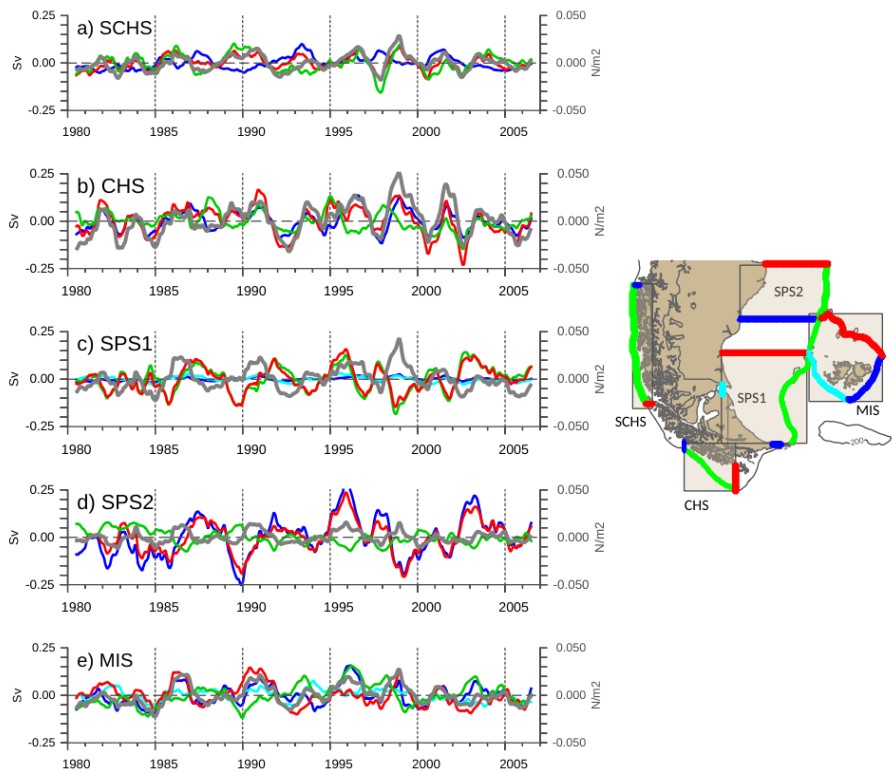

**Figure 9.** Monthly transport anomalies across the sections of the key areas. From top to bottom: SCHS (a), CHS (b), SPS1 (c), SPS2 (d), MIS (e). On the right, a reference map shows in colors the sections of each key area where the monthly transport has been calculated. The grey line in each panel is the zonal wind stress in the area, calculated in the shaded regions indicated in the reference map. A 12-month running mean is applied to the data to filter the seasonal signal.

anomalies entering the SPS1 from the shelf-break against climatic indices, for both simulations. The transport variations show a significant correlation with the Southern Annular Mode (SAM) index (Marshall, 2003), in both CMM and ORCA (respectively r = -0.52 and -0.63, Fig. 10)

To further investigate the relationship between the SAM and the shelf transport we carried out an EOF analysis of the zonal
and meridional winds based on the ERA-Interim data between 1980 and 2006. Though the mean zonal wind is significantly larger than the meridional wind, the amplitude of the interannual variability of both components is similar (Supplementary Fig. S1). The first EOF mode of the meridional wind (EOF1v), which explains 30% of the total variability, presents a dipole structure with positive loadings over most of the southern portion of the shelf around southern South America and negative loadings farther north in the Pacific and farther northeast in the Atlantic (Fig. 11 (c)). This spatial pattern is robust and also
emerges in the first EOF modes of sea level pressure and meridional wind in the Southern Hemisphere south of 20°S (see Supplementary Fig. S2). The time series of EOF1v presents substantial interannual variability, which is significantly correlated




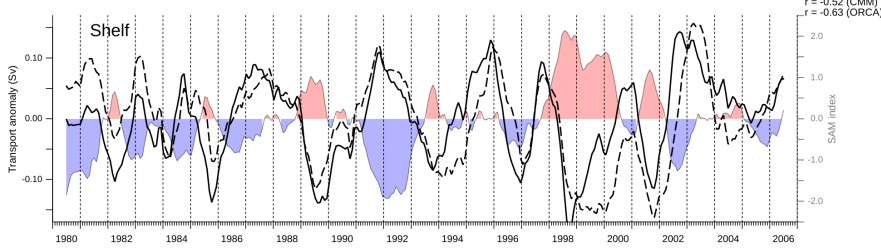

**Figure 10.** Panel A: Monthly transport anomaly through the SPS1 shelf-break sections from CMM (solid line) and ORCA (dashed line) and SAM index (positive/negative phases in red/blue). Correlations (r) between simulations and the SAM index are shown in the upper right-hand side. A 12-month running mean is applied to the data to filter the seasonal signal.

with the SAM index (-0.67, Fig. 11 (e)) and the northward transport through SPS1 (0.68, Fig. 11 (f)). These results are in agreement with the recent analyses of Combes and Matano (2018), who suggested that the wind stress modulates the magnitude of the cross-shelf exchanges at the southern boundary of the Atlantic shelf. As will be discussed below, the on-phase variations

of the meridional winds at either side of South America have significant dynamical implications for the circulation over the continental shelves. The first EOF mode of the zonal wind (EOF1u), which explains 38% of the total variability, shows a spatial structure very similar to EOF1v over southern South America, with a positive zonal band that reaches $40°$S (Fig. 11 (a)). Its time series is also moderately correlated with the SPS1 transport anomalies but they are out of phase, that is during positive SAM phases stronger westerly winds drives weaker northward shelf transport. Note that although stronger westerlies promote

larger northward Ekman transport they also generate larger (southward) geostrophic coastal currents (Palma et al., 2008) that are in phase with the shelf transport anomalies.

## 5   Conclusions

Our analyses of numerical simulations allow the study of the interannual variability of the exchange between the southern portions of the HLME and PLME and their variability over three decades. Results from two numerical models are used to

quantify the transport over the shelf. Both simulations have been validated based on observations and have been widely used (Combes and Matano, 2018; Duchez et al., 2014). We used model outputs featuring the same space and time resolutions (monthly outputs at $1/12°$). However the models were independently developed and feature different parametrizations. Though our objective is not to make a formal inter-comparison of the models, due to the scarcity of observations, a discussion of their similarities and differences is useful to qualitatively determine which features of the circulation and their variability are

more robust. While ORCA is a global configuration based on the NEMO code, the CMM configuration is a two-way nested simulation embedded in a $1/4°$ parent grid. Thus, elements of the large-scale circulation, such as the ACC, are resolved at $1/12°$ at a circumpolar scale in ORCA whereas it is simulated at this resolution only in the child-grid domain in CMM. The choice of the vertical coordinate system (40 sigma-levels in CMM vs 75 z-levels in ORCA) plays an important role in the


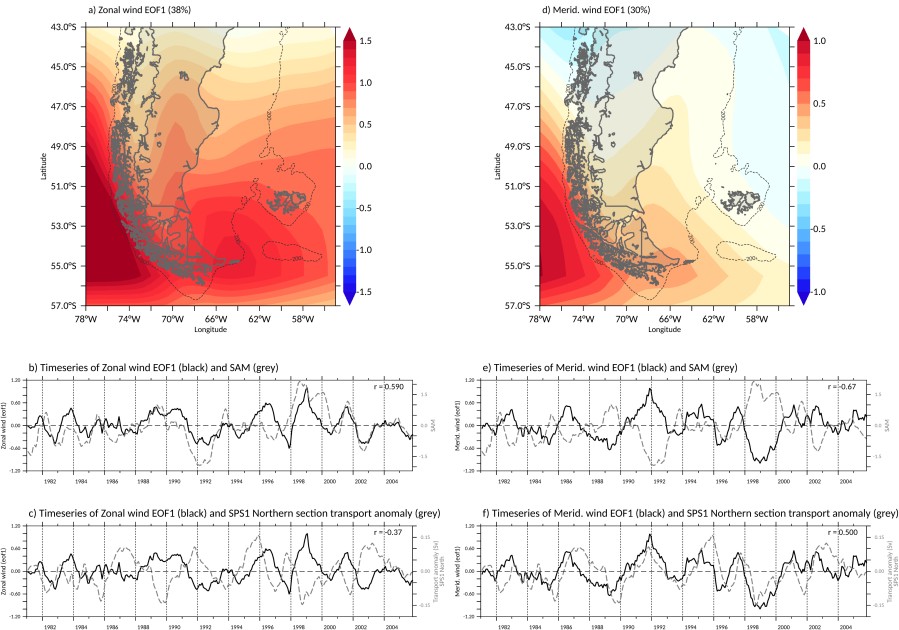

**Figure 11.** Panel a: Spatial pattern of first EOF of zonal winds around southern South America. panel b: Annual means of first EOF mode of zonal wind (black) and SAM index (grey) and (c) annual means of first EOF of zonal wind (black) and northward transport through the northern boundary of SPS1 (grey). (e) same as (b) but for the first EOF mode of the meridional wind. (f) same as (c) but for the first EOF mode of the meridional wind. A 12-month running mean is applied to the data.

representation of the shelf and slope circulation. ORCA features only 31 levels in the upper 200m. It is also important to note
that the Magellan strait inflow to the Atlantic shelf in the CMM simulation is implemented with a prescribed low salinity near the Atlantic mouth. In ORCA there is no correction on the MS salinity and therefore its discharge on the PS have salinities lower than observed. This point is of major importance in terms of water mass characteristics, but the resulting along-shelf transport is not significantly impacted by this correction, as the transport through the MS is similar in both simulations and its contribution in the overall mass balance of the southern PS is relatively small. Despite all these differences, both models are in
very good agreement in terms of circulation features over the shelf, and present comparable transport intensities and variability, though ORCA presents less intense shelf-open ocean exchange. For example, the transports across the shelf-break from 51 to $54.8°S$ (SPS1) in both models are well correlated (r = 0.75). This transport is also highly correlated with the SAM index in both simulations, suggesting that the wind variability partially controls the exchange between the shelf and the adjacent deep ocean. It is also important to note at this point that both simulations are forced at the surface by ERA reanalysis (ERA-40 for ORCA
and ERA-Interim for CMM). ERA-Interim features an improvement in resolution, data assimilation methods and physics but these changes do not impact greatly on the representation of the winds at monthly scale (Dee et al., 2011).

The combination of lagrangian and time series analysis allows better understanding of the pathways and quantifying the inputs of the different entry points into the southwest Atlantic shelf, and their variability. The shelf-deep ocean transports were





calculated across the 200-m isobath and not across strait sections, allowing a fine evaluation of the shelf-sea exchange. The

narrow SCHS is subject to the open ocean dynamics and the CHC, which flows along the shelf break. The PS waters originate from the SCHS and flow along the CHS. The majority of these waters enter the Atlantic shelf via Le Maire Strait and the ME shelf-break. An additional contribution through the MS, although smaller, is known to have a strong impact on the salinity and coastal circulation of the PS (Palma and Matano, 2012; Brun et al., 2019). The low-frequency variability of the shelf exchange in the SPS is partly explained by the large-scale wind variability, as determined by the SAM index.

The SAM modulates both wind components around southern South America. During positive SAM phases the zonal (meridional) component increases (decreases) and viceversa. The wind contribution on the shelf circulation however is known to be larger for along shelf (i.e. meridional) winds than for cross-shelf (zonal) winds (Greenberg et al., 1997). Therefore we hypothesize that the interannual variability of the along shelf transport over this region is largely associated with the variability of the meridional component of the wind, which is negatively correlated with the SAM index. We argue that variations in

the meridional component of the wind induce changes in the cross-shore Ekman transport, and set up a cross shore pressure gradient which in turn modulates the intensity of the along shore currents over the Pacific and Atlantic shelves. Interestingly, the first EOF mode of the meridional winds is uniform over the southern tip of South America, implying that wind anomalies are in-phase, on the Atlantic and Pacific shelves. During negative SAM index periods southerly winds prevail on both shelves, implying upwelling winds on the Pacific and downwelling winds on the Atlantic. Thus, according to the proposed mechanism,

the associated along shelf geostrophic transport anomalies in the southeast Pacific shelf must be $180°$ out-of-phase with the anomalies in the southwest Atlantic shelf (recall that positive transport is southward in Pacific shelf and northward in the Atlantic shelf). Such out-of-phase relationship is readily apparent when comparing the variability of the southward flow anomalies through the southern boundary of SCHS (black line in Fig. 12 (a)) and the transport anomalies through the shelf-break of SPS1 (r = -0.52, grey line in Fig. 12 (a)). Moreover, since the on-phase variability of the meridional wind over the Pacific and Atlantic

shelves implies out-of-phase variations in coastal sea level, there must be a direct impact on the barotropic pressure gradient between the Pacific and Atlantic mouths of the MS and hence on the associated transport. During positive SAM and negative (northerly) wind periods coastal sea level will rise in the Pacific and drop in the Atlantic, leading to increased transport from the Pacific to the Atlantic through the Magellan Strait. Such relationship is also evident in the strong correlation between the transport through the Atlantic mouth of the MS and the southward transport thorough the southern boundary of SCHS (r = 0.77,

Fig. 12 (b)). These results are consistent with the sensitivity analysis carried out on realistic simulations of the MS throughflow, which suggest a strong relationship between the inter-ocean sea level difference and the MS transport (Sassi and Palma, 2006). Given that the MS throughflow is the source of lowest salinity waters to the Atlantic shelf (Brun et al., 2019), periods of positive (negative) SAM, associated with enhanced (decreased) Magellan transport and decreased (enhanced) northeastward transport over the SPS should lead to low (high) salinity anomalies in this region. The SAM presents a positive long-term trend

in response to the increase of anthropogenic forcing (Marshall et al., 2006). Should this positive SAM trend persist in the future it would lead to significant changes in the water mass characteristics (lower salinity) of the southern PS.

The numerical simulations provide a regional understanding of the exchange between the two large marine ecosystems, and the availability of 3 decades of model outputs allows assessing the variability of this exchange. Studies at shorter time-scale





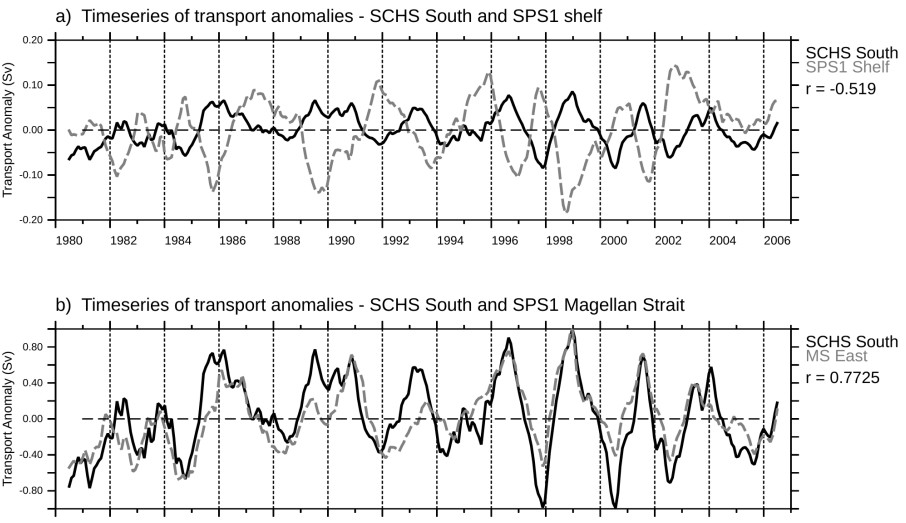

**Figure 12.** Panel a: Annual mean transport anomalies through the southern boundary of SCHS (black) and northward transport through SPS1 (gray). Panel b: Annual mean transport anomalies through the southern boundary of SCHS (black) and Pacific to Atlantic transport through the Atlantic mouth of the Magellan Strait (gray). The transports in b have been normalized with the maximum of each time series to help visualize the variability of the Magellan throughflow. The correlation between the transport anomalies is indicated on the left of each panel. A 12-month running mean is applied to the data to filter the seasonal signal.

and higher spatio-temporal resolution should be conducted in the future, focused on the $50 - 58°$ S latitude band, incorporating
continental discharge from glaciers and a better representation of the coastline and bottom topography. Model studies should
be carried out in conjunction with observations at key locations such as the Magellan Strait, Cape Horn Shelf, Le Maire Strait
and the shelf break along the Malvinas Embayment. This would improve the understanding of the water mass pathways in this
complex region.

*Author contributions.*  K. Guihou and E. D. Palma gathered the datasets. K. Guihou did the transport calculations. All authors participated in
the scientific interpretation of the results. K. Guihou prepared the manuscript with contributions from all co-authors.

*Competing interests.*  The authors declare that they have no conflict of interest.

*Acknowledgements.*  Karen Guihou was supported by a postdoctoral fellowship from Consejo Nacional de Investigaciones Científicas y
Técnicas (Argentina). This research was supported by the Inter-American Institute for Global Change Research grant number CRN3070,





which is financed by the US National Science Foundation grant GEO-1128040. The CMM data were facilitated by V. Combes from the
Oregon State University, USA, and the ORCA data by J. Harle, from the National Oceanography Center, UK.





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
