# Peer review of "Dynamical Connections between Large Marine Ecosystems of Austral South America based on numerical simulations"

_Ocean Science, 2019_

## Referee Comment (RC1) · Anonymous Referee #1 · 23 Oct 2019

The manuscript titled "Dynamical Connections between Large Marine Ecosystems of Austral South America based on numerical simulations" by Karen Guihou, Alberto R. Piola, Elbio D. Palma, and Maria Paz Chidichimo presents the analysis of a high resolution (1/12°) simulation of the area previously done by Combes and Matano (2014a), which is compared with the results from another (1/12°) numerical simulation. The connectivity between the Humboldt and the Patagonia Large Marine Ecosystems (HLME and PLME, respectively) are studied using the lagrangian tool ARIANE (Blanke and Reynauld, 1997) by calculating the transport across several defined sections.

This is a very interesting article that addresses an area of research that needs more

understanding, the connection between both sides of the South American southern coast. There is a very good design of the numerical experiments with ARIANE, and the analysis is details.

Minor issues:

a) L107-109: It is described that the domain extends to 81°W. However, Fig. 2 has its western boundary at 78°W. Please mention this detail to the reader.

b) L119: artic – should be Artic

Major issues:

A) It is stated that the Drake Passage and Cape Horn Shelf (CHS in manuscript) represent a key region (L212). However, both models have a limited representation of other pathways, namely the Magellan Strait and the Beagle Channel, and thus their impact cannot be understood with the models used. This is mentioned in L389-393, but please discuss further this issue in your analysis and describe a (numerical) solution to address it (not only "higher spatio-temporal resolution" L389).

B) L166-L67: state that the models used are "in good overall agreement" while several of the mean values from Table 1 show a quite large difference between them, particularly for the South Chilean Shelf (SCHS), which, it seems, is poorly represented by both models due to their grid size (1/12°), understanding that a model's effective resolution is larger (7-10 dx). Please moderate this optimistic statement comment on the issue.

C) Table 1 Shows mean, std, min and max values for the calculated transport. I would prefer to see the compared histograms of the calculated transport in both model. This would help to understand if the probability distribution function of the obtained volumes is gaussian and, thus, is adequate to use mean and std values as a statistical descriptors. Authors could also express the agreement for different percentiles for purposes of comparison-

---

## Referee Comment (RC2) · Anonymous Referee #2 · 25 Oct 2019

The manuscript analyses Eulerian transports and Lagrangian pathways between ecosystem regions on either side of southern South America. It is not my region of expertise, but from the past research described in the manuscript, it seems that previously there has been only limited observations and modelling work focused on this region. Therefore it appears that the results will fill a gap in the regional oceanographic knowledge and are worth publishing, after addressing the below comments.

Specific comments:

- Figure 1: It's not clear what the extent of the LMEs are from this map, can you add shading or an outline to show how large they are?

[Figure]

- line 35-36, "The large-scale circulation of the HLME includes the broad eastward flowing West Wind Drift at ≈ 43âŮȩS". This is a very outdated view of the Pacific Ocean circulation. See Chaigneau and Pizarro 2005 for updated terminology. The term "West Wind Drift" is no longer used by physical oceanographers, and the flow in the south-east Pacific covers the breadth of the gyre rather than being concentrated at 43âŮȩS as suggested here.

- line 42: "In this region". It's not clear what region you mean by this. I assume you are talking about the CHS, since that was mentioned in the last sentence. But since this is the start of a new paragraph, I wonder if you could be talking about the whole of the HLME again? But in the last paragraph you said that extended north to 4âŮȩS, where the westerlies are definitely not strong. Please clarify.

- line 42-44, "the main flow patterns are from the Pacific towards the Atlantic", this sentence needs a reference.

- lines 48-53: "Recent observational records with unprecedented spatial and temporal resolution in Drake Passage yielded an absolute ACC transport of 173.3±8.9Sv" etc. I am not sure how this level of detail is relevant, since your model transports are only measuring very close to the continent and in the upper ocean layers.

- model description at line 96: State how long the spinup is before the 27 years used here.

- line 120: What do you do with particles at the surface? Do you parameterize vertical motion within the surface mixed layer at all?

- there are so many acronyms used throughout the text for place names that are very difficult to follow and remember. E.g. line 135-137 ("Only a branch of this strong current, deflected between the PS and BB enters into the ME and flows along the Southern slope before joining the northwards flowing MC further north.") is ridiculous! I suggest spelling out some of the less familiar or less used terms, such as SCHS, CHS,

SPS, MIS, BB, PS throughout the manuscript.

Line 139: "vertically integrated up to 200-m deep". I think you mean integrated DOWN to 200m deep? Section 3.1: It would be good to provide some discussion here of how these transports compare with observations where possible. You listed so many transport values in the introduction where it wasn't obvious why the reader needed to know at that point. Move them here instead to validate you model circulation.

Line 175: I don't see what the point of these climatological Lagrangian experiments is. Why use a 1/12deg model if you don't include the mesoscale variability in your Lagrangian pathways? Past studies have shown that pathways are completely changed if you include the mesoscale variability. Later on you release at many different times and then average all of the subsequent trajectories to obtain an average picture of the pathways. Therefore I don't see what this initial analysis adds, since it is incorrect to leave out the time-varying flow. Alternatively, you could clearly state that this section is just to see what impact the mean flow, with no mescoscale variability, has on the pathways, and then later do a quantitative comparison of advection by the mean flow and advection by the complete time-varying velocity field. You say here that you want to "qualitatively investigate" the pathways, but including the mesoscale variability will actually give a different (correct) qualitative picture.

Line 217: What is the impact of only using monthly resolution velocity fields on your results?

References A. Chaigneau, O. Pizarro, Surface circulation and fronts of the South Pacific Ocean, east of 120°W, Geophysical Research Letters, 32, L08605 (2005).

---

## Referee Comment (RC3) · Anonymous Referee #3 · 6 Nov 2019

This study uses ROMS / NEMO ocean circulation models and a Lagrangian particle tracking routine to study transport of water around southern South America (1) between the deep ocean to coastal regions and (2) and between the Pacific and Atlantic coasts. The study represents a considerable amount of effort on the part of the authors and they have done a good job thoroughly exploring the models. I think the paper could be benefited by a few additional considerations.

In the Introduction the authors discuss the importance of this region from ecological, economic (fisheries) and climate perspectives. They argue that a better understanding of the ocean circulation in this region would be useful, particularly in the context of the

movement of larval/juvenile fishes. However, the authors never really come back to this idea in the Discussion or Conclusions. It would be helpful to know how their findings are useful in that regard.

Additionally, I wonder how sensitive your results are to use of monthly averaged velocity fields. An earlier study examining the sensitivity of transport predictions to the spatial and temporal resolution of ocean circulation model output found quite large differences between daily snapshots and monthly averages at 1/12 (0.08) degree resolution (https://royalsocietypublishing.org/doi/pdf/10.1098/rsif.2012.0979). For instance, the daily snapshots better represented the movement of oceanographic drifters and the 30 day averages tended to over-predict offshore transport (movement from the continental shelf to oceanic waters), but the overall distances traveled were reduced. Though most of the focus of that paper was in a western boundary current region, effects were also apparent in an eastern boundary current. I would feel more confident about the results if the transport predictions were either compared to some in situ Lagrangian measurement of ocean movement (e.g., drifters from NOAA's Global Drifter Database that contains 30+ years of drifter tracks, https://www.aoml.noaa.gov/phod/gdp/index.php) and/or to compare results to model output with daily (or better) temporal resolution. For this type of analysis you could consider only a subset of the years/results to compare. In my view all you need to do is to identify for the reader what type of bias (if any) they are looking at.

---

## Author Comment (AC1) · 5 Dec 2019

**Response to Referee # 1**

The manuscript titled "Dynamical Connections between Large Marine Ecosystems of Austral South America based on numerical simulations" by Karen Guihou, Alberto R. Piola, Elbio D. Palma, and Maria Paz Chidichimo presents the analysis of a high resolution (1/12) simulation of the area previously done by Combes and Matano (2014a), which is compared with the results from another (1/12) numerical simulation. The connectivity between the Humboldt and the Patagonia Large Marine Ecosystems (HLME and PLME, respectively) are studied using the lagrangian tool ARIANE (Blanke and Reynauld, 1997) by calculating the transport across several defined sections. This is a very interesting article that addresses an area of research that needs more understanding, the connection between both sides of the South American southern coast. There is a very good design of the numerical experiments with ARIANE, and the analysis is details.

1. **a) L107-109: It is described that the domain extends to 81W. However, Fig. 2 has its western boundary at 78W. Please mention this detail to the reader. b) L119: artic – should be Artic**
   Thanks for the observations. The manuscript has been modified accordingly.

2. **A) It is stated that the Drake Passage and Cape Horn Shelf (CHS in manuscript) represent a key region (L212). However, both models have**

**a limited representation of other pathways, namely the Magellan Strait and the Beagle Channel, and thus their impact cannot be understood with the models used. This is mentioned in L389-393, but please discuss further this issue in your analysis and describe a (numerical) solution to address it (not only "higher spatio-temporal resolution" L389).**

Although some geomorphological details of Magellan Strait (i.e., western sector, Primera and Segunda Angostura narrows) are crudely represented at 1/12 spatial resolution, transport estimates from CMM are very close to ORCA and the idealized and realistic models of Sassi and Palma (2006). These values are also in good agreement with recent hydrographic estimates (Brun et al, 2019, Estuarine, Coastal and Shelf Science, in revision). Beagle Channel is a very narrow passage ($\approx$ 5km at its narrowest point) connecting the SCHS and the CHS to the south of Tierra del Fuego island that is not represented in CMM nor in ORCA. Information about the magnitude and direction of the ocean currents or the mean transport inside this Channel is presently lacking. A recent analysis based on hydrographic data (Brun et al, 2019, in revision) suggests that though very low salinities are observed at certain locations within the Beagle Channel (e.g. Aguirre et al., Mar. Biol. Res., 2012), these highly diluted waters do not make a noticeable impact on the inter-ocean salinity exchange because they mix with saltier waters before reaching the Le Maire Strait. To properly address the possible importance of this Channel in the interoceanic exchanges we would need to consider a new model with higher resolution, focused on the 50 - 58S latitude band, incorporating continental discharge from glaciers and rivers and a better representation of the coastline and bottom topography (possibly using a nested subdomain or moving to unstructured grids like FVCOM). This effort should be carried out in conjunction with observations at key locations of the channel to properly incorporate the inflow/outflow from the outer ocean and validate model results.

The paragraph addressing this topic has been revised to discuss more extensively which kind of further simulations are needed (page 23 , line 409).

*The analysis of 27 years of numerical simulations provided a regional under-standing of the long-term exchange between the two large marine ecosystems, and allow assessing the variability of this exchange. Further studies at increased spatio-temporal resolution are clearly needed at some key locations that are under-resolved or absent in CMM and ORCA. In particular, efforts must be made to provide a better representation of the narrowest geomorphological features such as the western sector and the Primera and Segunda Angostura narrows of the Magellan Strait, and the Beagle Channel, a very long and narrow passage ( 5km) connecting the SCHS and the CHS to the south of Tierra del Fuego island. This task would require the development of a new regional model at higher resolution possibly nested to the 1/12 models or employing unstructured grids. The model should incorporate continental discharge from glaciers and rivers and a better representation of the coastline and bottom topography. At a temporal scale, daily outputs are required to improve the analysis of mesoscale variability, and evaluate their impact on the exchanges described in this study. The combination of such higher spatio-temporal simulations with the current simulations would provide a finer picture of the exchange, and allow quantification of the input from both the small-scale and large-scale circulation. Moreover, such modelling effort should be carried out in conjunction with observations at key locations of the Magellan Strait, Le Maire Strait, Beagle Channel, Cape Horn Shelf and the Malvinas Embayment, which would greatly improve our understanding of the water mass pathways in this complex region.*

3. **B) L166-L67: state that the models used are "in good overall agreement" while several of the mean values from Table 1 show a quite large difference between them, particularly for the South Chilean Shelf (SCHS), which, it seems, is poorly represented by both models due to their grid size (1/12), understanding that a model's effective resolution is larger (7-10 dx). Please moderate this optimistic statement comment on the issue.**

The sentence was corrected in the revised version.

4. **C) Table 1 Shows mean, std, min and max values for the calculated transport. I would prefer to see the compared histograms of the calculated transport in both model. This would help to understand if the probability distribution function of the obtained volumes is gaussian and, thus, is adequate to use mean and std values as a statistical descriptors. Authors could also express the agreement for different percentiles for purposes of comparison** Following the reviewer's advice, we computed the transport histograms (figure 1 attached, new Fig.3 in the revised version) and modified lines 165-173 of the manuscript accordingly (page 6, line 162).

*The above described transports are based on the numerical results from CMM. For comparison we have also analyzed the ORCA outputs over the same subregions. Fig. 3 displays the probability distribution of the monthly transport values for both models. The distributions are close to gaussian, the exception being very narrow pathways like the SCHS, the northwestern limit of the CHS, Magellan Strait and Le Maire Strait, especially in ORCA. Main differences between model results are concentrated on the southern sector of the SCHS where ORCA shows lower mean values in cross-shelf and alongshelf transports and decreased temporal variability. Additionally, and in contrast with the CMM results, the main source of waters to SPS1 in ORCA is the Le Maire Strait (0.52Sv), whereas only 0.35Sv enter via the shelf-break. In wider sectors like the CHS, the SPS1 and SPS2 (not shown) the agreement is better both for the mean and the variability.*

**References:**

Aguirre, GE., Capitanio, FL, Lovrich, GA, Esnal, GB. 2012. Seasonal variability of metazooplankton in coastal sub-Antarctic waters (Beagle Channel), Marine Biology

Research; 8; 4; 5-2012; 341-353.

Brun, A.A., et al., 2019. The role of the Magellan Strait on the southwest South Atlantic shelf. Estuar Coast Shelf Sci, in revision.

Sassi, M.G, Palma, E.D., 2006. Modelo Hidrodinámico del Estrecho de Magallanes; Asociación Argentina de Mecánica Computacional; Mec Comput, XXV; 16; 11; 1461-1477

**Supplement:**

[Figure]

**Figure 1** – *Histogram of transport over the 27 years, at key locations of a) the SCHS, b) the CHS, c) the SPS1, and d) the Malvinas Embayment, calculated along the colored sections indicated in the reference map. Opaque bins are from CMM, transparent bins from ORCA. Mean and standard deviation are indicated for each section. The transport is calculated over 200m for all sections except for the Malvinas Embayment sections, where it is computed throughout the water column.*

---

## Author Comment (AC2) · 5 Dec 2019

**Response to Referee # 2**

The manuscript analyses Eulerian transports and Lagrangian pathways between ecosystem regions on either side of southern South America. It is not my region of expertise, but from the past research described in the manuscript, it seems that previously there has been only limited observations and modelling work focused on this region. Therefore it appears that the results will fill a gap in the regional oceanographic knowledge and are worth publishing, after addressing the below comments.

1.  **Figure 1: It's not clear what the extent of the LMEs are from this map, can you add shading or an outline to show how large they are?**
    The extent of the HLME and PLME are defined in Heileman et al., 2009a/b, and the southern portions are basically bounded by the Cape Horn Current and Malvinas Current. We did not wish to outline the LMEs in figure 1 to prevent it from becoming too busy, but the references to these papers have been given in the revised manuscript.

2.  **line 35-36, "The large-scale circulation of the HLME includes the broad eastward flowing West Wind Drift at 43S. This is a very outdated view of the Pacific Ocean circulation. See Chaigneau and Pizarro 2005 for updated terminology. The term "West Wind Drift" is no longer used by physical oceanographers, and the flow in the south-east Pacific covers the breadth**
**of the gyre rather than being concentrated at 43S, as suggested here.**
Corrected (page 2, line 34)

*The HLME is a prototypical eastern boundary upwelling system that extends from northern Peru to southern Chile in the South Pacific Ocean (see Heileman et al., 2009 for a detailed description), where it is adjacent to the PLME (Fig. 1). The HLME can be separated in two meridional sub-regions, marked by the Subtropical Front, at 35 - 40S (Chaigneau and Pizarro, 2005). About 65% of the area of HLME corresponds to the northern region, and is under the influence of the Humboldt Current System and coastal upwelling from 4S to 40S. South of 45°S the HLME is mostly under the influence of downwelling-favorable (poleward) winds and the poleward flowing Cape Horn Current (CHC, Strub et al., 1998) along the shelf break of the Southern Chilean Shelf (SCHS, 40 - 55S), a region with a complex fjord system. Further south, the shelf widens onto The Cape Horn Shelf region (CHS), marking the northern boundary of the Drake Passage.*

3. **line 42: "In this region". It's not clear what region you mean by this. I assume you are talking about the CHS, since that was mentioned in the last sentence. But since this is the start of a new paragraph, I wonder if you could be talking about the whole of the HLME again? But in the last paragraph you said that extended north to 4S, where the westerlies are definitely not strong. Please clarify.**
The text has been clarified in the revised version.

4. **line 42-44, "the main flow patterns are from the Pacific towards the Atlantic", this sentence needs a reference.**
The reference was included in the revised version (Combes and Matano; 2014).

5. **lines 48-53: "Recent observational records with unprecedented spatial and temporal resolution in Drake Passage yielded an absolute ACC transport of 173.3+-8.9Sv" etc. I am not sure how this level of detail is relevant, since**

**your model transports are only measuring very close to the continent and in the upper ocean layers.**
Corrected. Lines 48-53 have been deleted in the revised version.

6.  **model description at line 96: State how long the spinup is before the 27 years used here.**
    The spin-up span 15 years, 10 years for the parent model and 5 additional years for the parent/child configuration. It was followed by a 34-year integration (1979–2012; Combes and Matano, 2014b). The text has been modified accordingly in the revised version.

7.  **line 120: What do you do with particles at the surface? Do you parameterize vertical motion within the surface mixed layer at all?**
    Particles are advected by the 3D simulated currents, meaning that surface particles are allowed to upwell/downwell if there is a vertical velocity component.

8.  **there are so many acronyms used throughout the text for place names that are very difficult to follow and remember. E.g. line 135-137 ("Only a branch of this strong current, deflected between the PS and BB enters into the ME and flows along the Southern slope before joining the northwards flowing MC further north.") is ridiculous! I suggest spelling out some of the less familiar or less used terms, such as SCHS, CHS, SPS, MIS, BB, PS throughout the manuscript.**
    The manuscript has been modified spelling out some of the less frequently used terms (i.e, BB, ME).

9.  **Line 139: "vertically integrated up to 200-m deep". I think you mean integrated DOWN to 200m deep?**
    Yes. Corrected.

10. **Section 3.1: It would be good to provide some discussion here of how**

**these transports compare with observations where possible. You listed so many transport values in the introduction where it wasn't obvious why the reader needed to know at that point. Move them here instead to validate you model circulation.**

The only existing measurements of transport (to the best of our knowledge) are close to the Drake Passage. In this section we focus on the mean transport on the shelf, for which we do not have observations. The best we can do is to compare our results with indirect transport estimates (Brun et al, 2019) or previous numerical studies (Palma et al, 2008; Combes and Matano, 2018).

11. **Line 175: I don't see what the point of these climatological Lagrangian experiments is. Why use a 1/12deg model if you don't include the mesoscale variability in your Lagrangian pathways? Past studies have shown that pathways are completely changed if you include the mesoscale variability. Later on you release at many different times and then average all of the subsequent trajectories to obtain an average picture of the pathways. Therefore I don't see what this initial analysis adds, since it is incorrect to leave out the time-varying flow. Alternatively, you could clearly state that this section is just to see what impact the mean flow, with no mescoscale variability, has on the pathways, and then later do a quantitative comparison of advection by the mean flow and advection by the complete time-varying velocity field. You say here that you want to "qualitatively investigate" the pathways, but including the mesoscale variability will actually give a different (correct) qualitative picture.**

Following the reviewer's advice, we re-write Lines 175 and 214 (Lines 172 and 213 in the revised manuscript)

Line 175

*Two sets of Lagrangian experiments using Ariane were conducted to investigate the fate of HLME and PLME waters. The first set was intended to explore the*

*impact of the mean flow and employs the CMM's climatological velocity fields. Initially particles were released on all shelf grid points onshore from the 200-m isobath (see green areas in figures 3 a, 4a and 5 a), at the surface, 50m and 100m depths, and followed during 90 days.*
Line 214
*The second group employed monthly averaged currents from CMM and monthly releases to better quantify the effect of mesoscale and long-term variability on the pathway of particles flowing through the northern Drake Passage. Particles were released on all grid points from 0 to 200m at 68.1W (Cape Horn), from the coast (at 55.7S) to 58.5S, and tracked during one year. This simulation was repeated monthly from 1980 to 2005, that is 321 simulations of 365 days each.*

12. **Line 217: What is the impact of only using monthly resolution velocity fields on your results?**
Putman and He (2013) used HYCOM model outputs to explore resolution effects on particle tracking experiments in the North Atlantic: Comparison with oceanographic drifters released in the Gulf Stream System indicated that daily snapshots better represented the real trajectories while 30-day averages tended to under-predict speeds (although not direction). Numerical experiments of particle dispersal also indicated a bias towards higher cross-shelf transport when using low resolution model outputs. Overall, however, their results were very dependent on particle release location and the associated dynamical characteristics of the regional ocean circulation (i.e., releases in the South Atlantic Bight were scarcely affected by temporal resolution). In this regard, it is important to note that our geographical area differs from the one analyzed in Putman and He (2013). The SCHS is very narrow and located on an eastern boundary while the PS is extremely wide, is bounded by a very stable western boundary current and is forced by large tides (not included in HYCOM) and steady westerly winds. Altimetry analysis has shown that this is a region (particularly the shelf) of very

low eddy variability [see for example and Goñi et al, (2008), their Fig. 9 for the PS and Meredith (2016), his Fig.2, for the entire region]. Some of the drawbacks of poor sampling can be alleviated with multiple releases and long term tracking (Putman and He, 2013). The release of particles every 30 days (not just a single day) the long-term span of our tracking experiments and the ensemble average of the results (Fig. 6) ensures a more robust interpretation of particle dispersion between the HLME and the PLME. Therefore, although we expect some influence of high frequency input on our preliminary particle tracking results, the only way to properly asses and quantify this effect would be to perform a dedicated series of experiments, something that is beyond the scope of our study. A paragraph on these matters was included in page 12, line 242 of the revised manuscript.

---

## Author Comment (AC3) · 5 Dec 2019

**Response to Referee # 3**

This study uses ROMS / NEMO ocean circulation models and a Lagrangian particle tracking routine to study transport of water around southern South America (1) between the deep ocean to coastal regions and (2) between the Pacific and Atlantic coasts. The study represents a considerable amount of effort on the part of the authors and they have done a good job thoroughly exploring the models. I think the paper could be benefited by a few additional considerations.

1. **In the Introduction the authors discuss the importance of this region from ecological, economic (fisheries) and climate perspectives. They argue that a better understanding of the ocean circulation in this region would be useful, particularly in the context of the movement of larval/juvenile fishes. However, the authors never really come back to this idea in the Discussion or Conclusions. It would be helpful to know how their findings are useful in that regard.**
   Thanks for pointing out this omission. We now inserted the following statement at the end of Section 5 (line 423 in the revised manuscript):

   *Despite the significant morphological and dynamical differences between the southern Humboldt Current and Patagonia LMEs a number of biogeographical studies based on a variety of species and different methodology have suggested*

*that south of 43° S both regions share similar biological and environmental characteristics and belong to the Magelleanic Province (e.g. Boschi, 2000a, b; Sullivan Sealey and Bustamante, 1999; Spalding et al., 2007 and references therein). However, the causes of this relatively strong ecological connectivity are unknown. Our results suggest the connectivity is mediated by a well-defined flux from the Pacific to the Atlantic, which may facilitate the dispersion of holoplanktonic species and planktonic larvae of benthic species, as well as fish larvae towards the Atlantic.*

2. **Additionally, I wonder how sensitive your results are to use of monthly averaged velocity fields. An earlier study examining the sensitivity of transport predictions to the spatial and temporal resolution of ocean circulation model output found quite large differences between daily snapshots and monthly averages at 1/12 (0.08) degree resolution (https://royalsocietypublishing.org/doi/pdf/10.1098/rsif.2012.0979). For instance, the daily snapshots better represented the movement of oceanographic drifters and the 30day averages tended to over-predict offshore transport (movement from the continental shelf to oceanic waters), but the overall distances traveled were reduced. Though most of the focus of that paper was in a western boundary current region, effects were also apparent in an eastern boundary current. I would feel more confident about the results if the transport predictions were either compared to some in situ Lagrangian measurement of ocean movement (e.g., drifters from NOAA's Global Drifter Database that contains 30+ years of drifter tracks, https://www.aoml.noaa.gov/phod/gdp/index.php) and/or to compare results to model output with daily (or better) temporal resolution. For this type of analysis you could consider only a subset of the years/results to compare. In my view all you need to do is to identify for the reader what type of bias (if any) they are looking at.**

The reviewer poses an interesting question regarding the spatio-temporal resolution of the ocean model that we employed to feed the particle tracking algorithm. Putman and He (2013) used HYCOM model outputs to explore resolution effects on particle tracking experiments in the North Atlantic. Better results were obtained with 1/12 spatial resolution and one-day temporal sampling. The CMM output has in principle adequate spatial resolution but otherwise is only available at a maximum frequency of 10 days (10day-average). Note that tidal forcing precludes the use of daily snapshots (unless the temporal sampling is set at hourly values, something requiring huge storage facilities). Putman and He (2013) also indicate that the resolution of the model output required to appropriately simulate organism movement depends on the question being addressed and the dominant physical factors influencing ocean dynamics over the regions and timescales of interest. In this regard, it is important to note that our geographical area differs from the one analyzed in Putman and He (2013). The SCHS is very narrow and located on an eastern boundary while the PS is extremely wide, is bounded by a very stable western boundary current and is forced by large tides (not included in HYCOM) and steady westerly winds. Altimetry analysis has shown that this is a region (particularly the shelf) of very low eddy variability [see for example and Goñi et al, (2011), their Fig. 9 for the PS and Meredith (2016), his Fig.2, for the entire region). CMM maximum temporal resolution is 10-day average. We made additional exploratory experiments with CMM comparing trajectories forced with 30-day (30D) averages and 10-day (10D) averages. We selected years 1992 and 1998 which show two extremes of the SAM cycle (Fig. 10). Some indication of higher drifter velocities in 10D can be seen near the surface. The overall pattern of particle dispersal for 10D, however, is very similar to 30D both for SCHS and PS (see Fig. 1 to 4 attached). The reviewer also points to a comparison between real and computational float trajectories. A qualitative comparison was included in Fig. 6 (real trajectories were plotted in light green). Speed of virtual floats

for some selected trajectories around CHS are ≈10% less than oceanographic drifters (not shown). A more quantitative (statistical) analysis however is precluded by the small number of real floats present in our study area (many of them got stalled in the Chilean fjord system). Some of the drawbacks of poor sampling can be alleviated with multiple releases and long term tracking (Putman and He, 2013). The release of particles every 30 days (not just a single day) the long-term span of our tracking experiments and the ensemble average of the results (see Fig. 6 of the manuscript) ensures a more robust interpretation of particle dispersion between the HLME and the PLME. Therefore, although we expect a reduced influence of high frequency input on our preliminary particle tracking results, the only way to proper asses and quantify this effect would be to perform a dedicated series of experiments (including model-model intercomparison), something that is beyond the scope of our study.

Following the reviewer advice a paragraph on this matters was included in the revised manuscript (page 13, line 242):

*Putman and He (2013) used HYCOM model outputs to explore resolution effects on particle tracking experiments in the North Atlantic. Comparison with oceanographic drifters released in the Gulf Stream System indicated that daily snapshots better represented the real trajectories while 30-day averages tended to under-predict speeds (although not direction). Numerical experiments of particle dispersal also indicated a bias towards higher cross-shelf transport when using low resolution model outputs. Overall, however, their results were very dependent on particle release location and the associated dynamical characteristics of the regional ocean circulation (i.e., releases in the South Atlantic Bight were scarcely affected by temporal resolution). In this regard, it is important to note that our geographical area differs from the one analyzed in Putman and He (2013). The SCHS is very narrow and located on an eastern boundary while the PS is extremely wide, bounded by a very stable western boundary current*

*and is forced by large tides (not included in HYCOM) and steady westerly winds. Altimetry analysis has shown that this is a region (particularly the shelf) of very low eddy variability [see for example and Goñi et al, (2008), their Fig. 9 for the PS and Meredith (2016), his Fig.2, for the entire region]. CMM maximum temporal resolution is 10-day average. We made additional exploratory experiments comparing trajectories forced with 30-day (30D) averages and 10-day (10D) averages. Some indication of higher drifter velocities in 10D can be seen near the surface. The overall pattern of particle dispersal for 10D, however, is very similar to 30D both for SCHS and PS (not shown).*

*Surface drifters' trajectories, extracted from the Coriolis database (http://www.coriolis.eu.org/) were compared against the above described simulated trajectories (green lines in Fig. 6). Eighty-nine trajectories flowing on the HLME/PLME or through the northern Drake Passage, were recorded between 1980 and 2017. Among them, 33 drifters flowed onto the HLME or the PLME shelf. However, most drifters either ended washing ashore in the numerous fjords of southern Chile, or were released directly in the PLME, and therefore are not useful to portray the exchange between the two shelves. Only two drifters went through the Drake Passage and penetrated the PS briefly, before reaching the ME. Three additional drifters directly entered the ME after flowing through the Drake Passage. Even though the observational dataset is small and precludes a quantitative statistical analysis, the observed and simulated trajectories are in good qualitative agreement as no drifters flowing through the Drake Passage south of the SAF entered the embayment. This supports our conclusion about the Pacific origin of the PS water masses.*

*Some of the drawbacks of poor sampling can be alleviated with multiple releases and long term tracking (Putman and He, 2013). The release of particles every 30 days (not just a single day) the long-term span of our tracking experiments and the ensemble average of the results (Fig. 6) ensures a more robust interpretation of particle dispersion between the HLME and the PLME. Therefore, although*

*we expect some influence of high frequency input on our preliminary particle tracking results, the only way to proper asses and quantify this effect would be to perform a dedicated series of experiments, something that is beyond the scope of our study.*

**References**

Boschi, E. E.: Species of Decapod Crustaceans and their distribution in the American marine zoogeographic provinces. Revista de investigación y Desarrollo Pesquero, 13, 7-13, 2000a. Boschi, E. E.: Biodiversity of marine decapod brachyurans of the Americas. Journal of Crustacean Biology, 20, 337-342, 2000b.

Goñi, G. J., F. Bringas and P. N. DiNezio. Observed low frequency variability of the Brazil Current front. J. of Geophys. Res (Oceans), 116, C10037, 1-10, 2011. Meredith, M. P. Understanding the structure of changes in the Southern Ocean eddy field, Geophys. Res. Lett., 43, 5829–5832, 2016.

Putman NF, and He R.. Tracking the long-distance dispersal of marine organisms: sensitivity to ocean model resolution. J R Soc Interface, 10: 20120979. http://dx.doi.org/10.1098/rsif.2012.0979, 2013. Spalding, M. D., Fox, H. E., Allen, G. R., Davidson, N., Ferdaña, Z. A., Finlayson, M. A. X., ... & Martin, K. D. Marine ecoregions of the world: a bioregionalization of coastal and shelf areas. BioScience, 57, 573-583, 2007.

Sullivan Sealey, K. and G. Bustamante. Setting geographic priorities for marine conservation in Latin America and the Caribbean. The Nature Conservancy, Arlington, Virginia, 125pp, 1999.

**Supplement:**

[Figure]

**Figure 1** – *Lagrangian advection of particles during 90 days, released on the SCHS in January 1992. Panel a: advection calculated in 10-day mean fields from CMM, panel b: advection calculated in 30-day (1 month) mean fields from CMM. Particle trajectories are colored by depth (in m) for initial release at 0 (left), 50 (middle) and 100 (right) m (1 out of 20 trajectories shown). The number of particles (n) for each experiment is shown and the SAF is indicated in black.*

[Figure]

**Figure 2** – *Same a fig 1, but for a release in January 1998.*

[Figure]

**Figure 3** – *Lagrangian advection of particles during 90 days, released on the PS in January 1992. Panel a: advection calculated in 10-day mean fields from CMM, panel b: advection calculated in 30-day (1 month) mean fields from CMM. Particle trajectories are colored by depth (in m) for initial release at 0 (left), 50 (middle) and 100 (right) m (1 out of 20 trajectories shown). The number of particles (n) for each experiment is shown and the SAF is indicated in black.*

[Figure]

**Figure 4** – *Same a fig 4, but for a release in january 1998.*